# SampDetox: Black-box Backdoor Defense via Perturbation-based Sample Detoxification

**Yanxin Yang[1], Chentao Jia[1], DengKe Yan[1], Ming Hu[2] \*, Tianlin Li[3], Xiaofei Xie[2],**
**Xian Wei[1], Mingsong Chen[1] \***

[1]MoE Eng. Research Center of SW/HW Co-Design Tech. and App., East China Normal University
[2]Singapore Management University, [3]Nanyang Technological University
{52275902023, 51265902040, 51265902053}@stu.ecnu.edu.cn,
hu.ming.work@gmail.com, tianlin001@ntu.edu.sg, xfxie@smu.edu.sg,
{xwei, mschen}@sei.ecnu.edu.cn

## Abstract

The advancement of Machine Learning has enabled the widespread deployment of Machine Learning as a Service (MLaaS) applications. However, the untrustworthy nature of third-party ML services poses backdoor threats. Existing defenses in MLaaS are limited by their reliance on training samples or white-box model analysis, highlighting the need for a black-box backdoor purification method. In our paper, we attempt to use diffusion models for purification by introducing noise in a forward diffusion process to destroy backdoors and recover clean samples through a reverse generative process. However, since a higher noise also destroys the semantics of the original samples, it still results in a low restoration performance. To investigate the effectiveness of noise in eliminating different types of backdoors, we conducted a preliminary study, which demonstrates that backdoors with low visibility can be easily destroyed by lightweight noise and those with high visibility need to be destroyed by high noise but can be easily detected. Based on the study, we propose SampDetox, which strategically combines lightweight and high noise. SampDetox applies weak noise to eliminate low-visibility backdoors and compares the structural similarity between the recovered and original samples to localize high-visibility backdoors. Intensive noise is then applied to these localized areas, destroying the high-visibility backdoors while preserving global semantic information. As a result, detoxified samples can be used for inference even by poisoned models. Comprehensive experiments demonstrate the effectiveness of SampDetox in defending against various state-of-the-art backdoor attacks. The source code of this work is publicly available at https://github.com/easywood0204/SampDetox.

## 1 Introduction

With Artificial Intelligence (AI) technologies showing advantages in tasks such as image classification and target detection, they are widely used in safety- and security-critical applications such as autonomous processing [1, 2], IoT System [3, 4] and health-care [5, 6]. As Machine Learning as a Service (MLaaS) [7, 8, 9] becomes increasingly popular, AI applications are relying on AI services provided by third parties. However, since users cannot guarantee the trustworthiness of service providers, they face serious threats of backdoor attacks [10, 11, 12]. Typically, adversaries inject backdoors into deep models on numerous occasions, especially when the training samples are collected from unreliable sources, or the pre-trained deep models are obtained from untrusted third parties. During the inference phase, when fed clean samples, the backdoored models behave normally

---

*Ming Hu and Mingsong Chen are the corresponding authors.

38th Conference on Neural Information Processing Systems (NeurIPS 2024).

with satisfactory classification performance. However, when processing poisoned samples containing trigger patterns, the backdoored models will be fooled into predicting attack target categories with high confidence.

Although various defense methods have been proposed to identify and destroy backdoors, most of them are white-box defenses that require additional training samples or the right to analyze model parameters [13, 14, 15]. Since in MLaaS-based applications users are not allowed to access the original training data and parameters of models, these backdoor defense methods are strongly limited. In contrast, black-box backdoor defense methods do not have such requirements. So far, black-box backdoor defense methods can be classified into three categories, i.e., model detection-based [16], sample detection-based [7, 17], and sample purification-based [18, 19] defense methods. The basic idea of the first two categories is to simply discard training samples or deep models once they are identified as poisoned. In this case, the usability and performance of classification tasks are greatly affected. As an alternative, sample purification-based approaches strive to destroy backdoor trigger patterns in samples. However, existing sample purification-based approaches are based on the strong assumption that trigger patterns are small and only located in the corners of samples, which is not always true in practice. As a result, such methods can only be used to defend against specific types of backdoor attacks. Worse still, most of the above defense methods need to repetitively feed samples to deep models and get their continuous feedback, resulting in extra non-negligible computation overhead. *Therefore, how to effectively mitigate the impacts of all possible backdoor attacks without deteriorating the overall inference performance is becoming a great challenge in black-box defense.*

Intuitively, adding noise to poisoned samples can disrupt the semantics of the backdoor, thereby eliminating it from the sample. To preserve the semantics of the original sample while removing the backdoor, the diffusion model [20, 21] serves as an effective solution to restore the sample. However, different backdoors exhibit varying levels of robustness to noise, which requires different noise amplitudes for effective disruption. In addition, the amplitude and range of noise will affect the quality of samples restored by the diffusion model. We performed a preliminary study on various backdoor attacks to explore the robustness of different types of backdoors. We found that: i) invisible triggers with low robustness (to random noise) can be easily destroyed, and ii) backdoor triggers with high robustness are easily detectable due to their high visibility.

Inspired by these observations, to address the challenge of black-box backdoor defense, we propose a novel two-stage method called SampDetox. Based on our first observation, we globally apply coarse-granularity noises on each given sample in the first stage of SampDetox, where backdoor triggers with low visibility can be easily destroyed. To counteract the effects of such added noises, we then restore the samples by denoising them using diffusion models. Note that for a poisoned sample, its poisoned regions with high visibility, compared with clean regions, will be different from their counterparts in the restored version (see the proof in Theorem 4.2), making them identified easily. Based on our second observation, the second stage can quickly locate such robust triggers and destroy them using our dedicatedly designed noises. Similar to the first stage, the second stage then leverages diffusion models to restore the samples. Since our approach does not pose any assumptions or requirements on the models, it accommodates arbitrary backdoor attack scenarios. In summary, this paper makes the following three major contributions:

- We perform a preliminary study on a wide spectrum of backdoors to reveal the correlation between the visibility of triggers and the robustness of poisoned samples.
- We present a novel perturbation-based sample detoxification method together with its theoretical foundations. Our approach can effectively destroy all possible triggers with dedicatedly designed noises and does not compromise the overall inference performance.
- We conduct extensive experiments to show the applicability and superiority of our approach over state-of-the-art (SOTA) backdoor defense methods.

## 2 Related Work

**Backdoor Attacks.** Existing backdoor attacks can be mainly classified into two categories, i.e., visible attacks and invisible attacks, based on the visibility of trigger patterns during the inference phase. Specifically, visible attacks do not consider the concealment of trigger patterns. For example, BadNets [22] uses visible trigger patterns in the form of fixed pixels patched in the corner of the images. Although there exist various visible attacks (e.g., training set corruption without label

poisoning [23], label-consistent backdoor attacks [24], attacking pre-trained models [25], and specific-sample triggers [26]), few of them can be applied in real scenarios, since the triggers on samples are conspicuous and can be easily detected by manual inspection. To make attacks stealthier, more and more backdoor attacks (e.g., Blended [27], WaNet [28], and ISSBA [29]) adopt invisible triggers. For example, BPP [30] uses image quantization and dithering as imperceptible backdoor triggers to avoid manual inspection.

**Backdoor Defenses.** According to the capabilities of defenders in manipulating the training and inference processes on deep models, backdoor defense methods can be classified into two categories, i.e., white-box and black-box defenses. Typically, white-box methods defend against backdoor attacks by changing training processes [31, 32], extract gradients [33], and fine-tuning and -pruning of poisoned models [34, 35]. However, these methods require access to model parameters, which strongly restricts their usage (e.g., MLaaS applications) in practice. For black-box defenses, such as model detection-based [16], sample detection-based [7, 17], and sample purification-based [18, 19] approaches, defenders only can rely on the predictions of input samples. For example, as a model detection-based approach, CBD [36] can effectively identify whether models are poisoned by analyzing their predictions. However, simply discarding the suspicious model/sample does not apply to real-world classification tasks. Rather than ignoring poisoned samples, sample purification-based methods aim to destroy the triggers on samples. For example, Sancdifi [37] measures saliency maps and purifies the specific areas of samples. BDMAE [19] reconstructs the local area of the samples using its AutoEncoder. ZIP [18] transforms and reconstructs samples independently of models through zero-shot image purification. However, all these purification-based methods can handle only small triggers within specific regions.

The robustness of triggers against perturbations or modifications plays an important role in determining the performance of backdoor attacks and defenses. For example, CBD [36] analyzes the robustness of triggers with different sizes against random noises to certify its proposed defense scheme. However, CBD only investigates attacks with simple triggers without taking SOTA backdoor attacks into account. To the best of our knowledge, SampDetox is the first attempt to consider the correlation between the visibility of backdoor triggers and the robustness of poisoned samples. As a model-independent method, SampDetox enables perturbation-based sample detoxification, which can effectively defend against complex backdoor attacks within arbitrary application scenarios.

# 3 Preliminary Study

This section first reviews the process of backdoor attacks. Then, it gives our definitions of the visibility of backdoor triggers and the robustness of poisoned samples and reveals their inherent correlation based on a preliminary study.

## 3.1 Evaluation Metrics

Assume that $M$ is a model poisoned by some backdoor attack. Let $x^c$ be a clean sample, and $x^p$ represent its poisoned version by adding specific backdoor triggers. If the backdoor attack succeeds, the model $M$ should classify $x^c$ into its correct category while classifying $x^p$ into a target category. To defend against backdoor attacks, the findings in [38] show that transforming or perturbing samples will achieve defensive effects against locally patched triggers. However, according to the observations in [18, 19], these operations are unsuitable for defending against other types of triggers. This is mainly because locally patched triggers are unremarkable and susceptible to disturbance, while other triggers are robust (i.e., not sensitive) to such backdoor defenses. To better understand the inherent characteristics of backdoor attacks and the trends of their development, we conduct a preliminary study on existing attacks to reveal the correlation between the visibility of backdoor triggers and the robustness of their poisoned samples, whose descriptions are as follows.

**Visibility.** When a backdoor attack is applied to some deep model, the visibility of embedded triggers indicates the stealthiness of the backdoor attack. In this paper, we use the Structural Similarity Index Measure (SSIM) [29, 39] to evaluate the visibility of backdoor triggers. This is because, unlike traditional measurements like Mean Squared Error (MSE) and Peak Signal-to-Noise Ratio (PSNR), SSIM takes into account the local characteristics of samples, aligning more closely with the visual system of human eyes. Let $\text{SSIM}(x^c, x^p) \in [-1, 1]$ be the SSIM between $x^c$ and $x^p$, quantifying the visibility of trigger patterns embedded in $x^p$. For ease of evaluation, we adopt the notation

$v = (1 - \text{SSIM}(x^c, x^p))/2$ to normalize the visibility values. Specifically, the higher the value of $v$, the more notable the triggers are.

**Robustness.** For a given poisoned sample $x^p$, we propose to use the term robustness to quantify its capability to resist perturbations or modifications imposed by random noise. Assume that the backdoored model classifies $x^p$ into a specific attack target category due to the existence of triggers. We apply random noises to $x^p$ and get a modified sample $x_m$, where the modification process can be described by the equation $x_m = (1 - \eta_r)x^p + \eta_r\epsilon$ ($\epsilon \sim \mathcal{N}(0, \mathbf{I})$). Since $\eta_r$ plays an important role in determining whether its modified version $x_m$ belongs to another category, we use it to reflect the robustness of $x^p$. Typically, the higher the value of $\eta_r$, the more robust the poisoned sample $x^p$ is.

## 3.2 Observations from Attacks

To explore the correlation between the visibility of triggers and the robustness of their host poisoned samples, we conduct an experiment on the dataset CIFAR-10 against ten backdoor attacks, including both visible and invisible attacks, where each attack is applied on 200 randomly selected samples from CIFAR-10. Note that in the experiment, we assume that each poisoned sample is touched by only one kind of backdoor attack. In other words, each poisoned sample is embedded with a specific trigger. For each poisoned sample, we calculate the visibility score $v$ of the trigger and the robustness $\eta_r$ of the poisoned sample. As shown in Figure 1, for different poisoned counterparts of a clean sample, their visibility and robustness scores differ significantly under different backdoor attacks.

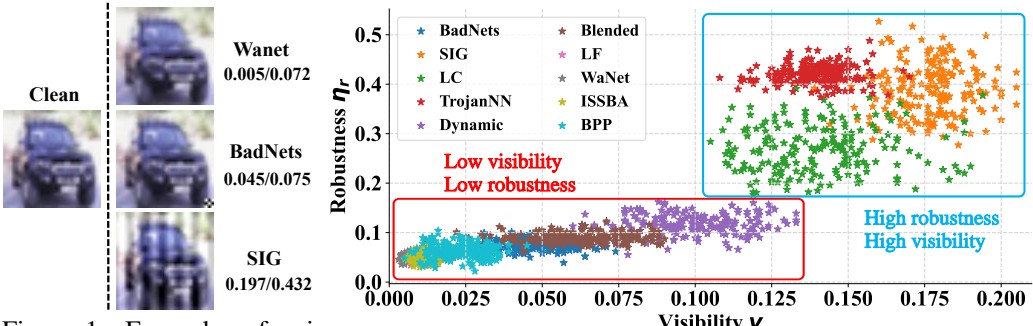

Figure 1: Examples of poisoned samples and their $v/\eta_r$.

Figure 2: Correlation between $v$ and $\eta_r$.

Figure 2 illustrates the $(v, \eta_r)$ pairs for all the poisoned samples under ten attacks (detailed in Section 5.1), where each filled pentagram symbol represents a poisoned sample. From the figure, we can get the following two observations.

**Observation 1.** For backdoor triggers with low visibility ($v < 0.13$), their host samples (i.e., ones with $\eta_r < 0.18$ located in the red box) have low robustness. According to [38], the triggers within these poisoned samples can be easily destroyed by lightweight noises.

**Observation 2.** For poisoned samples (i.e., those located in the blue box) with high robustness ($\eta_r \geq 0.18$), their backdoor triggers have high visibility, which renders them easily detectable and accurately located.

## 4 Our SampDetox Approach

### 4.1 Threat Model

We consider defending against backdoor attacks in black-box scenarios [17, 18], where defenders do not have access to the parameters of deep models and can only obtain predictions for samples. For attackers, we assume that they can completely control the training processes of models, enabling them to poison samples in datasets and modify model components to achieve poisoned models. In our approach, defenders strive to detoxify poisoned samples to classify them into their original correct categories. For clean samples, defenders aim to maintain their classification accuracy.

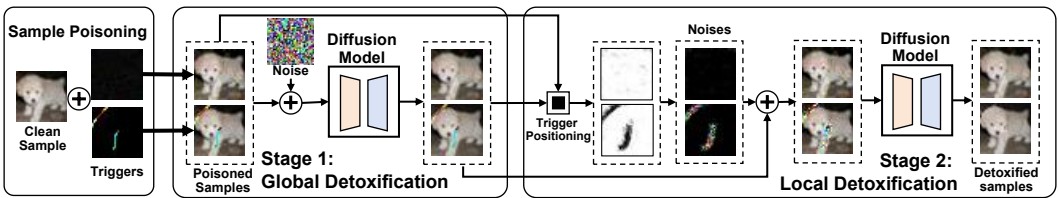

Figure 3: Framework and workflow of our SampDetox approach

## 4.2 Overview of SampDetox

As observed in Section 3.2, there exists a strong correlation between the visibility of triggers and the robustness of poisoned samples, which can be used for backdoor defense. However, due to the lack of prior knowledge of clean samples, it is difficult to figure out both the trigger visibility and robustness of poisoned samples. To defend various backdoor attacks, Figure 3 presents a novel perturbation-based backdoor defense framework using our proposed sample detoxification method, which consists of two stages:

**Stage 1: Global Detoxification.** According to Observation 1, backdoor triggers with low visibility typically have low robustness poisoned samples. In this case, such triggers can be easily destroyed by adding lightweight noises to suspected samples. However, due to the low visibility or even invisibility, it is hard to determine the existence or location of triggers within a given sample. To destroy all these potential triggers, Stage 1 applies the noise-based perturbation (see Section 4.3) globally to each sample using lightweight noises. Note that the introduction of noises inevitably affects the inference accuracy of deep models. To counteract the impacts of added noises on inference performance, we then utilize DDPM-based denoising (see Section 4.4) to denoise the samples. In this way, Stage 1 can destroy possible triggers spread over the poisoned samples with low robustness.

**Stage 2: Local Detoxification.** According to Observation 2, poisoned samples with high robustness are often embedded with highly visibility triggers. Based on DDPM-based denoising in Stage 1, the clean regions of poisoned samples can easily be restored to their original states, while the poisoned regions cannot, especially when triggers are highly visible. Due to this fact, by comparing a potentially poisoned sample with its detoxified version by Stage 1 pixel by pixel, we can figure out the positions of obvious triggers and the required intensity of local noises. Then, Stage 2 applies the noise-based perturbation locally on each sample using pixel-specific noises to destroy such triggers. Similarly to Stage 1, the added noises in Stage 2 also greatly influence the final inference accuracy for both clean and detoxified poisoned samples, thus requiring diffusion models to perform the denoising. After Stage 2, both visible and invisible triggers can be safely eliminated without degrading the inference performance.

## 4.3 Noise-based Perturbation

To quantitatively control the intensity of noises applied on samples in each stage of SampDetox, we resort to the forward Denoising Diffusion Probabilistic Model (DDPM) [20], which can also be used to determine the number of denoising steps required in the DDPM-based denoising process. Assume that $x_0$ is an input sample, and $x_i$ is the sample after adding noises for $i$ times (steps). In the forward process of DDPM, the model incrementally adds noises to $x_0$ step by step until the sample (denoted by $x_T$) is completely composed of random Gaussian noises at step $T$, where $T$ is a diffusion model-specific constant. In each step $t$, $x_t$ is obtained by adding a random Gaussian noise $\epsilon \sim \mathcal{N}(\mathbf{0}, \mathbf{I})$ to $x_{t-1}$ based on the formula $x_t = \sqrt{1 - \beta_t} x_{t-1} + \sqrt{\beta_t} \epsilon$, where $\beta_t \in (0, 1)$. For a given $x_{t-1}$, the posterior probability of $x_t$ can be represented as $q(x_t | x_{t-1}) = \mathcal{N}(x_t; \sqrt{1 - \beta_t} x_{t-1}, \beta_t \mathbf{I})$. Since step $t$ only relies on step $t$-1, the forward process can be regarded as a Markov process in the form of $q(x_t | x_0) = \prod_{t=1}^{T} q(x_t | x_{t-1})$. For a given $x_0$, we can get the probability of $x_t$ as

$$P(x_t | x_0) = \mathcal{N}(x_t; \sqrt{\overline{\alpha}_t} x_0, (1 - \overline{\alpha}_t) \mathbf{I}), \tag{1}$$

where $\alpha_t = 1 - \beta_t$ and $\overline{\alpha}_t = \prod_{i=1}^{t} (1 - \beta_t)$. According to [20], we can get the exact relationship between $x_0$ and $x_t$ using the formula

$$x_t = \sqrt{\overline{\alpha}_t} x_0 + \sqrt{1 - \overline{\alpha}_t} z, \tag{2}$$

where $z \sim \mathcal{N}(\mathbf{0}, \mathbf{I})$ is a random Gaussian noise.

Leveraging the forward DDPM, our approach aims to add proper random Gaussian noise on samples within $t$ steps based on Equation 2, which can destroy backdoor triggers in samples. Therefore, Theorem 4.1 investigates the potential of DDPM to eliminate triggers within poisoned samples.

**Theorem 4.1.** *Suppose that $x_t$ represents a diffused sample at step $t$ of the forward process in DDPM, where $t \in [0, T]$. Given a clean sample $x_0^c$ and its poisoned version $x_0^p$, we obtain $x_t^c$ and $x_t^p$ through the forward DDPM, whose distributions are $P_t(x)$ and $Q_t(x)$, respectively. By using the Kullback-Leibler divergence [40, 41] to measure the distance between the two distributions, we have*

$$KL(P_{t+1}||Q_{t+1}) \leq KL(P_t||Q_t), \tag{3}$$

*where the equality happens only when $P_t = Q_t$., and the inequality implies that the KL divergence between $P_t$ and $Q_t$ monotonically decreases along with the forward DDPM.*

Please refer to Appendix B.1 for the proof of the theorem. According to Theorem 4.1, for any positive number $\xi$, there exists a minimum $\bar{t} \in [0, T]$ such that for any $t \geq \bar{t}$ we have $KL(P_t||Q_t) \leq \xi$. Note that this theorem forms the theoretical basis to utilize the forward DDPM to minimize the distribution discrepancy between poisoned and clean samples, thus mitigating the impact of backdoor triggers.

## 4.4 DDPM-based Denoising

We also utilize DDPM to perform denoising on samples within the two stages of SampDetox. For each pixel in a sample, the number of denoising steps is determined by its noise-based perturbation steps (i.e., $\bar{t}$) as introduced in Section 4.3. In reverse DDPM, by taking the current sample state $x_t$ as an input, the diffusion model can be used to achieve its previous state $x_{t-1}$. Given both $x_0$ and $x_t$, according to Bayes' Theorem, we can obtain the probability distribution of $x_{t-1}$ as $p(x_{t-1}|x_t, x_0) = p(x_t|x_{t-1}) \cdot p(x_{t-1}|x_0)/p(x_t|x_0)$. By resorting to the relationship between $x_0$ and $x_t$ described by Equation 2, according to [20], we can use $x_t$ to approximate $x_0$ as follows:

$$x_0 = \frac{1}{\sqrt{\overline{\alpha}_t}}(x_t - \sqrt{1 - \overline{\alpha}_t}z_t), \ z_t = \theta(x_t, t), \tag{4}$$

where $z_t$ is predicted by the pre-trained neural network $\theta$ of DDPM, denoting the estimation of the real noise in step $t$. Accordingly, we can obtain $x_{t-1}$ from $x_t$:

$$x_{t-1} = \frac{1}{\sqrt{\alpha_t}}(x_t - \frac{1 - \alpha_t}{\sqrt{1 - \overline{\alpha}_t}}\theta(x_t, t)) + \sigma_t z, \tag{5}$$

where $\sigma_t^2 = \frac{1 - \overline{\alpha}_{t-1}}{1 - \overline{\alpha}_t} \cdot \beta_t$ and $z \sim \mathcal{N}(\mathbf{0}, \mathbf{I})$.

Unlike existing DDPM-based methods that strive to restore samples from Gaussian noises, SampDetox uses diffusion models to denoise samples generated by our noise-based perturbation approach (see Section 4.3). Assuming that the training data used for the pre-trained diffusion model do not involve any knowledge about backdoor triggers, according to the principle of DDPM, the denoised results will restore the original samples with all the triggers cleansed. Theorem 4.2 investigates the difference between clean and poisoned regions of poisoned samples after denoising.

**Theorem 4.2.** *Suppose that $R(x)$ denotes a specific region of the sample $x$. Given a poisoned sample $x_0^p$, we use $R_c(x_0^p)$ and $R_p(x_0^p)$ to represent a clean region and a poisoned region of the sample, respectively. We perform the noise-based perturbation on $x_0^p$ for $\bar{t}$ steps and get the noisy samples $x_{\bar{t}}^p$, and then utilize DDPM to get a denoised sample $\widehat{x}_0^p$. When comparing different regions of the original sample $x_0^p$ with their counterparts of the detoxified sample $\widehat{x}_0^p$, we have*

$$\frac{||R_p(\widehat{x}_0^p) - R_p(x_0^p)||^2}{n_1} > \frac{||R_c(\widehat{x}_0^p) - R_c(x_0^p)||^2}{n_2}, \tag{6}$$

*where $n_1$ and $n_2$ represent the numbers of pixels in $R_p(x_0^p)$ and $R_c(x_0^p)$, respectively.*

Please refer to Appendix B.2 for the proof of the theorem. According to Theorem 4.2, we can calculate the similarity between the denoised and original sample to identify the positions of trigger patterns. The inequality in Theorem 4.2 implies that for a poisoned sample, its poisoned regions, compared with clean regions, will be more different from their counterparts in the restored version.

## 4.5 Implementation of SampDetox

Algorithm 1 details the implementation of SampDetox. Lines 1-6 describe the process of global detoxification, while Lines 8-21 present the local detoxification process. Specifically, Line 1 applies the noise-based perturbation to the sample following Equation 2 with a fixed noise intensity $\bar{t}_1$. Lines 2-6 show the process of DDPM-based denoising to obtain the denoised sample $\hat{x}_0$. In Line 7, SampDetox calculates the SSIM score for each pixel between the original sample $x_0$ and its denoised version $\hat{x}_0$. Lines 8-11 describe how a noise with a specific intensity (i.e., $m[i]$) is added to the pixel $i$. Lines 12-21 present the DDPM-based denoising process, whether SampDetox iteratively calculates the estimation of the real noise $z_t$ with $x_t$ using the diffusion model $\theta$ and then selectively denoises each pixel accordingly. In Lines 16-19, SampDetox only updates pixels with noise intensity higher than that of the current step. Finally, Line 22 returns the resulting detoxified sample $x_0$, which is the denoised sample by Stage 2.

---

**Algorithm 1** Implementation of SampDetox

**Input:** i) $x_0$, original input sample; ii) $\bar{t}_1, \bar{t}_2$, the numbers of noise-adding and denoising steps in Stage 1 and Stage 2, respectively; iii) $\theta$, diffusion model;

**Output:** a detoxified sample;

1: $x_{\bar{t}_1} \leftarrow \sqrt{\alpha_{\bar{t}_1}} x_0 + \sqrt{1 - \alpha_{\bar{t}_1}} z$  $//z \sim \mathcal{N}(\mathbf{0}, \mathbf{I})$
2: **for** $t = \bar{t}_1$ to 1 **do**
3:     $z \sim \mathcal{N}(\mathbf{0}, \mathbf{I})$ if $t > 1$, else $z = 0$
4:     $\sigma_t^2 \leftarrow \frac{1 - \bar{\alpha}_{t-1}}{1 - \bar{\alpha}_t} \cdot \beta_t$
5:     $x_{t-1} \leftarrow \frac{1}{\sqrt{\alpha_t}}(x_t - \frac{1 - \alpha_t}{\sqrt{1 - \bar{\alpha}_t}} \theta(x_t, t)) + \sigma_t z$
6: **end for**
7: $M \leftarrow \text{PixelSSIM}(x_0, \hat{x}_0)$
8: **for** each pixel $i$ in $M$ **do**
9:     $m[i] \leftarrow \bar{t}_2 \cdot M[i]$
10:     $x_{\bar{t}_2} \leftarrow \sqrt{\alpha_{m[i]}} \cdot \hat{x}_0 + \sqrt{1 - \alpha_{m[i]}} z$  $//z \sim \mathcal{N}(\mathbf{0}, \mathbf{I})$
11: **end for**
12: **for** $t = \bar{t}_2$ to 1 **do**
13:     $z \sim \mathcal{N}(\mathbf{0}, \mathbf{I})$ if $t > 1$, else $z = 0$
14:     $z_t \leftarrow \theta(x_t, t)$
15:     **for** each pixel $i$ in $M$ **do**
16:         **if** $m[i] > t$ **then**
17:             $x_{t-1}[i] \leftarrow \frac{1}{\sqrt{\alpha_t}}(x_t[i] - \frac{1 - \alpha_t}{\sqrt{1 - \bar{\alpha}_t}} z_t[i]) + \sigma_t z[i]$
18:             **else** $x_{t-1}[i] \leftarrow x_t[i]$
19:         **end if**
20:     **end for**
21: **end for**
22: **return** $x_0$

---

# 5 Experiments

To evaluate the effectiveness of our approach, we implemented our approach, i.e., SampDetox, on top of Pytorch (version 1.13.0). We compared the defense performance between SampDetox and state-of-the-art (SOTA) black-box defense methods against various well-known backdoor attacks. All experiments were carried out on an Ubuntu workstation equipped with one Intel i7-13700K CPU, 64GB memory, and one NVIDIA GeForce RTX4090 GPU.

## 5.1 Experimental Setup

**Dataset and Model Settings.** We investigated three classical datasets (i.e., CIFAR-10 [42], GT-SRB [43], and Tiny-ImageNet [44]) and three models (i.e., PreAct-ResNet18, ResNet34 [45] and VGG-19 [46]). Due to space limitations, this section only presents the experimental results of PreAct-ResNet18 on the CIFAR-10 dataset. Note that we can find similar trends from the experiments using different models and datasets. Please refer to Appendix C for more details.

**Evaluation Metrics.** To objectively evaluate the effectiveness of a given backdoor defense method, we adopted the following three metrics: i) Clean sample Accuracy (CA), which denotes the inference accuracy of clean samples processed by the defense method; ii) Poisoned sample Accuracy (PA), which indicates the inference accuracy of poisoned samples purified (detoxified) by the defense method based on their ground-truth labels; and iii) Attack Success Rate (ASR), which represents the rate of successful attacks by poisoned samples. Note that a higher CA means that the defense method causes less interference in the classification of clean samples. Typically, backdoor defenders strive to achieve both high CA and PA with lowered ASR.

**Attack Methods.** We conducted defenses against ten SOTA backdoor attacks, i.e., BadNets [22], SIG [23], Label Consistent (LC) [24], TrojanNN [25], Dynamic [26], Blended [27], Low Frequency (LF) [47], WaNet [28], ISSBA [29], and BPP [30]. Note that the first five are visible attacks, while the last five are invisible attacks. We used the same benchmark settings in [48] to configure all these attack methods. Please refer to Appendix A.1 for more details.

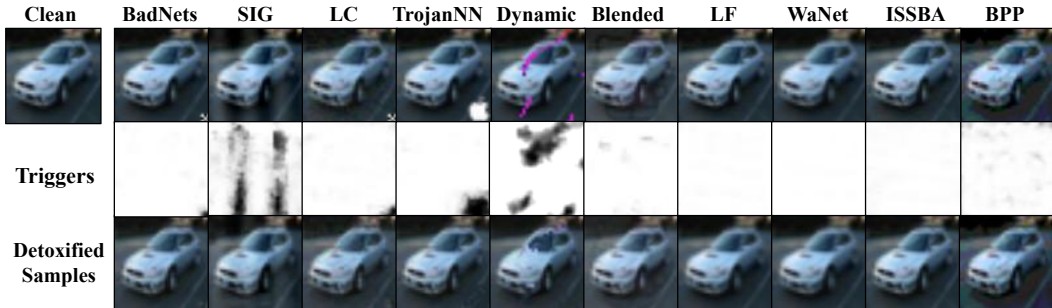

Figure 4: Illustration of poisoned samples, backdoor triggers, and their corresponding detoxified samples for the ten attacks, respectively.

Table 1: Defense performance comparison, where **Bold** fonts and underlines denote the best and the second-best values, respectively.

| Defense→ | No Defense | | | Sancdifi | | | BDMAE | | | ZIP | | | SampDetox (Ours) | | |
|---|---|---|---|---|---|---|---|---|---|---|---|---|---|---|---|
| Attack↓ | CA(%) | PA(%) | ASR(%) | CA(%) | PA(%) | ASR(%) | CA(%) | PA(%) | ASR(%) | CA(%) | PA(%) | ASR(%) | CA(%) | PA(%) | ASR(%) |
| No Attack | 93.84 | - | - | - | - | - | - | - | - | - | - | - | - | - | - |
| BadNets | 92.00 | 10.18 | 99.97 | 76.59 | 89.55 | **1.92** | 89.02 | 90.10 | 2.32 | 88.12 | 86.52 | 7.17 | **89.57** | **90.15** | 2.11 |
| SIG | 84.94 | 9.78 | 98.50 | 70.68 | 43.73 | 29.58 | 82.77 | 10.08 | 96.65 | 82.15 | 35.60 | 36.58 | **83.71** | **65.06** | **11.03** |
| LC | 84.34 | 10.26 | 99.06 | 68.44 | 51.78 | 3.64 | 79.75 | 73.39 | 2.01 | 79.85 | **74.92** | 2.06 | **80.72** | 74.36 | **1.55** |
| TrojanNN | 93.20 | 11.07 | 99.03 | 76.63 | **90.84** | **1.81** | 91.19 | 89.35 | 2.47 | 87.35 | 86.91 | 7.10 | **92.78** | 89.95 | 1.86 |
| Dynamic | 91.09 | 10.02 | 98.19 | 76.24 | 68.89 | 7.92 | 88.48 | 75.78 | 12.57 | 87.96 | 80.19 | 2.75 | **88.52** | **88.62** | **1.45** |
| Blended | 93.85 | 10.93 | 99.51 | 77.92 | 51.12 | 15.06 | 87.84 | 14.88 | 96.46 | 88.51 | 63.81 | 8.72 | **90.23** | **86.65** | **1.96** |
| LF | 93.63 | 11.13 | 99.48 | 77.92 | 49.95 | 16.57 | 87.50 | 13.63 | 80.97 | 88.76 | 86.59 | 5.85 | **90.01** | **87.40** | **3.02** |
| WaNet | 91.43 | 10.27 | 91.05 | 77.87 | 42.97 | 14.35 | 85.95 | 23.19 | 50.63 | 86.91 | 85.22 | 8.36 | **89.34** | **88.92** | **5.59** |
| ISSBA | 93.57 | 11.38 | 95.96 | 77.70 | 52.05 | 14.20 | 86.18 | 53.12 | 22.39 | 87.75 | 85.46 | 1.79 | **90.74** | **86.51** | **1.60** |
| BPP | 91.38 | 9.46 | 98.40 | 75.32 | 50.42 | 15.25 | 86.69 | 21.73 | 53.46 | 85.42 | 82.94 | 7.20 | **90.59** | **84.83** | **6.15** |

**Defense Baselines.** We compared SampDetox with three SOTA sample purification-based backdoor defenses: i) Sancdifi [37], which utilizes the RISE algorithm to measure the saliency maps and purify the specific areas of samples; ii) BDMAE [19], which uses a masked AutoEncoder to defend against backdoor attacks; and iii) ZIP [18], which applies linear transformations to samples and restores them under noises by using diffusion models. Note that both Sancdifi and BDMAE need to feed samples into deep models repeatedly to obtain their predictions, while ZIP does not rely on predictions. Please refer to Appendix A.2 for more details.

### 5.2 Performance Comparison

Figure 4 shows an example of backdoor defense against the ten different attacks using our proposed SampDetox. Here, the first row presents both a clean sample and all its poisoned counterparts. The second row presents the triggers identified by stage 2 of SampDetox, while the third row exhibits the detoxified samples of each attack. From this figure, we can find that SampDetox can effectively figure out all visible and invisible triggers embedded by different SOTA backdoor attacks.

Table 1 compares the defense performance between SampDetox and the three baselines on the CIFAR-10 dataset using PreAct-ResNet18. Please see Appendix C.1 for the results of datasets GTSRB and Tiny-ImageNet. From this table, we can find that SampDetox can achieve the highest CA for all different attacks, indicating that the impact of SampDetox on clean samples is negligible. Meanwhile, SampDetox has the best PA and ASR for 8 out of 10 attacks, respectively. Even for all the remaining cases (i.e., BadNets, LC, and TrojanNN), our approach can achieve the second-best PA and ASR, whose values are quite close to the ones of the best PA and ASR. As an example of the BadNets attack, Sancdifi has the best ASR, which is only $0.19\%$ lower than our approach. However, in this case, our approach significantly outperforms Sancdifi in CA by $12.98\%$. For the TrojanNN attack, though Sancdifi can achieve slightly better PA (by $0.89\%$) and ASR (by $0.05\%$) than SampDetox, SampDetox notably outperforms Sancdifi in CA by $16.15\%$. For the Label Consistent (LC) attack, ZIP can achieve slightly better PA, but its CA and ASR are worse than SampDetox. It is important to note that Sancdifi and BDMAE require feedback from deep models. As an alternative, our approach does not rely on underlying models, showing the applicability and superiority of our approach.

Table 2: Ablation study results of our approach against backdoor attacks with different visibility.

| Attack | Visibility | Noise* | | | Stage 1 | | | SampDetox (Stage 1 + Stage 2) | | |
|---|---|---|---|---|---|---|---|---|---|---|
| | $v$ | CA(%) | PA(%) | ASR(%) | CA(%) | PA(%) | ASR(%) | CA(%) | PA(%) | ASR(%) |
| BadNets | 0.052 | 56.25 | 49.71 | 3.93 | 90.12 | 88.15 | 6.61 | 89.57 | 90.15 | 2.11 |
| SIG | 0.185 | 51.81 | 17.16 | 15.89 | 83.10 | 9.48 | 92.33 | 83.71 | 65.06 | 11.03 |
| LC | 0.121 | 47.75 | 28.04 | 1.07 | 81.50 | 61.45 | 34.75 | 80.72 | 74.36 | 1.55 |
| TrojanNN | 0.137 | 58.03 | 45.89 | 5.68 | 92.54 | 35.51 | 46.86 | 92.78 | 89.95 | 1.86 |
| Dynamic | 0.098 | 56.26 | 41.18 | 1.09 | 87.58 | 85.62 | 3.82 | 87.52 | 88.62 | 1.45 |
| Blended | 0.067 | 59.42 | 45.22 | 1.53 | 88.65 | 86.65 | 1.86 | 90.23 | 86.65 | 1.96 |
| LF | 0.005 | 56.12 | 40.95 | 2.55 | 89.43 | 87.50 | 3.02 | 90.01 | 87.40 | 3.02 |
| WaNet | 0.005 | 57.71 | 43.73 | 5.38 | 89.43 | 88.91 | 5.60 | 89.34 | 88.92 | 5.59 |
| ISSBA | 0.006 | 57.10 | 37.35 | 1.14 | 90.92 | 86.50 | 1.61 | 90.74 | 86.51 | 1.60 |
| BPP | 0.009 | 58.40 | 42.03 | 5.75 | 89.09 | 84.84 | 6.15 | 90.59 | 84.83 | 6.15 |

## 5.3 Ablation Studies

**Impacts of Different Stages and Denoising.** The implementation of SampDetox consists of two stages. The first stage strives to destroy triggers globally on a given sample in a coarse manner, while the second stage tries to identify fine triggers and remove them locally. To examine the impacts of these two stages, we consider a variant (i.e., "Stage 1") of our approach, which does not take stage 2 of SampDetox into account. Meanwhile, to evaluate the impact of the denoising operations, we considered another variant of SampDetox (i.e., "Noise*"), where denoising is not applied to restore the samples in both stages. Table 2 presents the ablation study results against the ten attacks. Note that to enable an intuitive understanding of the role of each stage in SampDetox, this table provides the average visibility of backdoor triggers for each attack.

For all the defense methods in Table 2, we can find that the smaller the trigger visibility caused by backdoor attacks, the lower PA and higher ASR we can achieve. Although "Noise*" can achieve comparable ASR results to SampDetox, SampDetox outperforms "Noise*" in both CA and PA significantly. It means that simply adding noises to samples can effectively destroy the backdoor triggers of the samples. However, the noises themselves will also result in a sharp decline in the inference accuracy of clean and poisoned samples. If we merely adopt the first stage of SampDetox due to the denoising operation by diffusion models, both CA and PA can be improved. However, since the first stage only coarsely applies the perturbation on samples, most subtle backdoor triggers still survive. As an example in SIG attack, "Stage 1" can only achieve a PA of $9.48\%$ but with an ASR of $92.33\%$. By combining both Stages 1 and 2, SampDetox can effectively identify such undetectable triggers and destroy them without affecting the inference of original samples, leading to a better PA and ASR. For example, in SIG attack, by using SampDetox, we can achieve a PA of $65.06\%$ and an ASR of $11.03\%$.

**Impact of Hyperparameters $\bar{t}_1$ and $\bar{t}_2$.**
In our approach, we use two hyperparameters, i.e., $\bar{t}_1$ and $\bar{t}_2$ to control the noise intensity and denoising samples, respectively. We investigated two variants of SampDetox to evaluate the impacts of $\bar{t}_1$ and $\bar{t}_2$. For the first variant, we fixed the value of $\bar{t}_2$ (i.e., $\bar{t}_2$=0) while allowing the tuning of $\bar{t}_1$. With different values of $\bar{t}_1$, we can evaluate the defense performance of SampDetox against various invisible backdoor attacks (i.e., Blended, Low Frequency,

Table 3: Ablation study results for different $\bar{t}_1$ and $\bar{t}_2$.

| Fixed $\bar{t}_2 = 0$ | | | | Fixed $\bar{t}_1 = 20$ | | | |
|---|---|---|---|---|---|---|---|
| $\bar{t}_1$ | CA(%) | PA(%) | ASR(%) | $\bar{t}_2$ | CA(%) | PA(%) | ASR(%) |
| 5 | 92.07 | 62.48 | 27.58 | 40 | 92.19 | 60.90 | 30.07 |
| 10 | 91.22 | 78.69 | 12.96 | 60 | 92.02 | 73.32 | 19.32 |
| 15 | 90.92 | 86.35 | 5.39 | 80 | 91.86 | 79.38 | 14.11 |
| 20 | 90.65 | 86.43 | 1.73 | 100 | 91.72 | 83.68 | 6.13 |
| 25 | 88.26 | 85.13 | 1.80 | 120 | 92.02 | 85.22 | 2.34 |
| 30 | 86.77 | 84.56 | 1.71 | 150 | 91.85 | 84.92 | 2.28 |
| 35 | 84.91 | 83.78 | 1.75 | 200 | 92.26 | 81.87 | 2.30 |
| 40 | 82.30 | 83.01 | 1.72 | 250 | 92.39 | 77.03 | 2.29 |

WaNet, ISSBA, and BPP). Similarly, for the second variant, we fixed the value of $\bar{t}_1$ (i.e., $\bar{t}_1$=20) while changing the values of $\bar{t}_2$, to evaluate the defense performance of SampDetox against visible backdoor attacks (i.e., BadNets, SIG, LC, TrojanNN, and Dynamic). Table 3 shows the ablation study results with varying values of $\bar{t}_1$ or $\bar{t}_2$. Note that the results in the table represent the average results across multiple attack types. From this table, we can observe that when $\bar{t}_1$ and $\bar{t}_2$ increase, the ASR decreases, indicating the effect of adding noises in destroying backdoor triggers. However, a high value of $\bar{t}_1$ and $\bar{t}_2$ will also lead to a significant reduction of CA and PA, showing that the denoising processes are strongly affected by excessive noises. Therefore, we suggest to set $\bar{t}_1 = 20$ and $\bar{t}_2 = 120$ in practice.

## 5.4 Discussion

To show the applicability of SampDetox, we investigated the extra time overheads, explored training sample detoxification, evaluated the defense performance of SampDetox against SOTA adaptive attacks, and analyzed its limitations. Due to the space limitation, this section only presents the overhead caused by inference detoxification. Please refer to Appendix D for more discussions.

**Extra time overhead.** Based on our proposed two-stage perturbation-based sample detoxification, SampDetox can protect any deep models during their inference phase. However, similar to other backdoor defense approaches, the detoxification process will inevitably lead to extra time overhead. To address this issue, SampDetox resorts to the Denoising Diffusion Implicit Model (DDIM) [21] to accelerate the denoising process. Figure 5 compares the average inference time per sample for different back-

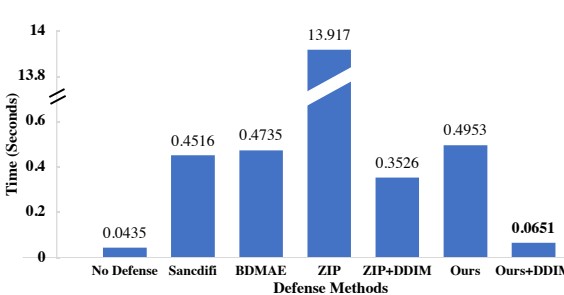

Figure 5: Inference time of different defense methods.

door defense methods. We can find that with the help of DDIM, the inference time of SampDetox is comparable to that of the method without defense.

## 6 Conclusion

Based on our observed correlation between the visibility of backdoor triggers and the robustness of poisoned samples, this paper introduces a novel black-box backdoor defense approach named SampDetox. By using our proposed perturbation-based sample detoxification, SampDetox can easily destroy backdoor triggers in samples and restore the samples using diffusion models. Extensive experiments against various complex backdoor attacks show the superiority of SampDetox from both perspectives of effectiveness and applicability to arbitrary scenarios.

## Acknowledgement

This work was supported by the Natural Science Foundation of China (62272170), "Digital Silk Road" Shanghai International Joint Lab of Trustworthy Intelligent Software (22510750100), Shanghai Trusted Industry Internet Software Collaborative Innovation Center, the National Research Foundation, Singapore, and the Cyber Security Agency under its National Cybersecurity R&D Programme (NCRP25-P04-TAICeN). Any opinions, findings and conclusions or recommendations expressed in this material are those of the author(s) and do not reflect the views of National Research Foundation, Singapore and Cyber Security Agency of Singapore.

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

# A    Detailed Experimental Setup

## A.1    Attack Method Settings

**BadNets** [22]. BadNets is the first proposed backdoor attack method, which patches a fixed pattern on the corners of benign samples and modifies their labels to the attack target category. We use a $3 \times 3$ black and white checkerboard in the lower right corner of samples as the trigger pattern.

**SIG** [23]. SIG modifies the pixels in samples based on the sinusoidal signal to generate poisoned samples. We set the frequency of the sinusoidal signal to 3 as the trigger pattern.

**Label Consistent (LC)** [24]. LC generates the poisoned samples using adversarial attack methods and keeps the original sample label unchanged. The trigger pattern is identical to BadNets, and other settings follow the original paper.

**TrojanNN** [25]. TrojanNN injects backdoors into a pre-trained model instead of injecting backdoors during the training. We use a white apple image as the trigger pattern.

**Dynamic** [26]. Dynamic trains a trigger generator to generate the specific triggers for each sample. The backdoor attack will only succeed if the specific trigger matches its sample. We follow the settings in the original paper to generate triggers.

**Blended** [27]. Blended generates poisoned samples by blending benign samples with a fixed pattern. Blending with coefficient $\alpha$, Blended increases the invisibility of the trigger to avoid manual inspection. We use a "hello kitty" image as the trigger pattern.

**Low Frequency (LF)** [47]. LF proposes an optimization-based approach to filter high-frequency artifacts to generate smooth triggers. We follow the settings in the original paper to generate triggers.

**WaNet** [28]. WaNet uses fixed warping functions to distort benign samples slightly to construct the poisoned samples. We follow the settings in the original paper to generate poisoned samples with trigger patterns.

**ISSBA** [29]. ISSBA employs image steganography to embed the string trigger into benign samples, which generates unique triggers that are specific to each sample. Following the settings in the original paper, we use the trained image steganography method.

**BPP** [30]. BPP generates the poisoned samples using image quantization. To inject the backdoor more efficiently, BPP proposes a contrastive learning-based approach to improve the success rate of backdoor attacks. We set the negative ratio to 0.1.

## A.2    Defense Baseline Settings

**Sancdifi** [37]. Sancdifi followed the settings in Section "Numerical Experiment" of its paper. Sancdifi computed RISE maps using 2000 random binary masks and set the saliency threshold to 0.95. For a fair comparison, Sancdifi used the same diffusion models as ours.

**BDMAE** [19]. BDMAE followed the settings in Section "Method Configurations" of its paper, which used masked autoencoders pre-trained on ImageNet with 24 encoder layers.

**ZIP** [18]. ZIP followed the settings in Section "Purification Implementation" of its paper, which set its hyperparameter to 2. For a fair comparison, ZIP used the same diffusion models as ours.

## A.3    Diffusion Model Settings

We utilize the framework of the improved diffusion model provided by [21]. The number of total diffusion steps was set to 1000 for all the datasets in the experiments, the noise_schedule was set as "cosine", and the learning rate was set to 1e-4. In the experiments, we load the parameters of the pre-trained model for each dataset. For DDIM, compared with the total steps of the diffusion model (e.g., $T = 1000$), our approach does not require too many steps to add noise and denoise samples (e.g., $\bar{t}_1 = 20$ and $\bar{t}_2 = 120$). Therefore, when we use DDIM to accelerate the denoising, we set the speed-up pace to 20 to ensure the effectiveness of our approach.

### A.4 PixelSSIM Settings

In our work, the PixelSSIM denotes an intermediate data structure for calculating the SSIM [39] value of two given samples. Specifically, for each block pair in the two samples, we calculate its SSIM value and record it in PixelSSIM at the same position as one of the central pixels of the block pair. When sliding the block pair along the two dimensions of the samples, we can collect all the SSIM values for the block pairs and form the final PixelSSIM. Note that by averaging all the values in PixelSSIM, we can achieve the SSIM value for the two samples. We use $structural\_similarity()$ in the $skimage$ library of Python to implement the PixelSSIM.

## B  Theoretical Justification

### B.1  Proof of Theorem 4.1

**Theorem 4.1.** *Suppose that $x_t$ represents a diffused sample at step $t$ of the forward process in DDPM, where $t \in [0, T]$. Given a clean sample $x_0^c$ and its poisoned version $x_0^p$, we obtain $x_t^c$ and $x_t^p$ through the forward DDPM, whose distributions are $P_t(x)$ and $Q_t(x)$, respectively. By using the Kullback-Leibler divergence [40, 41] to measure the distance between the two distributions, we have*

$$KL(P_{t+1}||Q_{t+1}) \leq KL(P_t||Q_t), \tag{7}$$

*where the equality happens only when $P_t = Q_t$, and the inequality implies that the KL divergence between $P_t$ and $Q_t$ monotonically decreases along with the forward DDPM.*

*Proof:* First, following the definition of the diffused sample probability distribution in DDPM [20], like Equation 1, when given a clean sample $x_0^c$ and a poisoned sample $x_0^p$, we have their distributions:

$$
\begin{aligned}
P(x_t^c|x_0^c) &= \mathcal{N}(x_t^c; \sqrt{\overline{\alpha}_t}x_0^c, (1 - \overline{\alpha}_t)\mathbf{I}) \\
Q(x_t^p|x_0^p) &= \mathcal{N}(x_t^p; \sqrt{\overline{\alpha}_t}x_0^p, (1 - \overline{\alpha}_t)\mathbf{I})
\end{aligned}
\tag{8}
$$

where $\alpha_t = 1 - \beta_t$ and $\overline{\alpha}_t = \prod_{i=1}^{t} \alpha_i$. Therefore, we know that both $P(x_t^c|x_0^c)$ and $Q(x_t^p|x_0^p)$ are Gaussian distributions. When both terms of the KL divergence are Gaussian distributions, we can simplify the formula as follows:

$$
\begin{aligned}
KL\left(\mathcal{N}\left(\mu_1, \sigma_1^2\right) \| \mathcal{N}(\mu_2, \sigma_2^2)\right) &= \int p(x) \log \frac{p(x)}{q(x)} dx \\
&= \int p(x) \left( \log \frac{\frac{1}{\sqrt{2\pi\sigma_1^2}} e^{-(x-\mu_1)^2/2\sigma_1^2}}{\frac{1}{\sqrt{2\pi\sigma_2^2}} e^{-(x-\mu_2)^2/2\sigma_2^2}} \right) dx \\
&= \int p(x) \log \left\{ \frac{\sqrt{\sigma_2^2}}{\sqrt{\sigma_1^2}} \exp \left\{ \frac{1}{2} \left[ \frac{(x-\mu_2)^2}{\sigma_2^2} - \frac{(x-\mu_1)^2}{\sigma_1^2} \right] \right\} \right\} dx \\
&= \frac{1}{2} \left( \log \sigma_2^2 - \log \sigma_1^2 + \int p(x) \frac{(x-\mu_2)^2}{\sigma_2^2} dx - 1 \right) \\
&= \frac{1}{2} \left( \log \sigma_2^2 - \log \sigma_1^2 + \frac{\mathbb{E}\left(x^2\right) - 2\mu_2 \mathbb{E}(x) + \mu_2^2}{\sigma_2^2} - 1 \right) \\
&= \frac{1}{2} \left( \log \sigma_2^2 - \log \sigma_1^2 + \frac{\sigma_1^2 + (\mu_1 - \mu_2)^2}{\sigma_2^2} - 1 \right)
\end{aligned}
\tag{9}
$$

Since $P_t$ and $Q_t$ have the same $\sigma$, i.e., $\sigma = \sigma_1 = \sigma_2 = \sqrt{1 - \overline{\alpha}_t}\mathbf{I}$, we can continue to simplify the formula:

$$
\begin{aligned}
KL(P_t||Q_t) &= \frac{1}{2} \left( \frac{(\mu_1 - \mu_2)^2}{\sigma_2^2} \right) \\
&= \frac{1}{2} \left( \frac{\overline{\alpha}_t(x_0^c - x_0^p)^2}{1 - \overline{\alpha}_t} \right)
\end{aligned}
\tag{10}
$$

Finally, we take the difference of both sides of the inequality

$$
\begin{aligned}
KL(P_{t+1}||Q_{t+1}) - KL(P_t||Q_t) =& \frac{1}{2}\left(\frac{\overline{\alpha}_{t+1}(x_0^c - x_0^p)^2}{1 - \overline{\alpha}_{t+1}}\right) - \frac{1}{2}\left(\frac{\overline{\alpha}_t(x_0^c - x_0^p)^2}{1 - \overline{\alpha}_t}\right) \\
=& \frac{(x_0^c - x_0^p)^2}{2}\left(\frac{\overline{\alpha}_{t+1}}{1 - \overline{\alpha}_{t+1}} - \frac{\overline{\alpha}_t}{1 - \overline{\alpha}_t}\right) \\
=& \frac{(x_0^c - x_0^p)^2}{2}\left(\frac{\overline{\alpha}_t\alpha_{t+1}}{1 - \overline{\alpha}_t\alpha_{t+1}} - \frac{\overline{\alpha}_t}{1 - \overline{\alpha}_t}\right) \\
=& \frac{(x_0^c - x_0^p)^2}{2}\left(\frac{1}{1 - \overline{\alpha}_t\alpha_{t+1}} - \frac{1}{1 - \overline{\alpha}_t}\right)
\end{aligned}
\tag{11}
$$

Since $\alpha_t = 1 - \beta_t$ and $\beta_t \in (0,1)$, $\alpha_t$ should be less than 1. As a result, we can get

$$
KL(P_{t+1}||Q_{t+1}) - KL(P_t||Q_t) \le 0. \tag{12}
$$

According to Equation 12, we have

$$
KL(P_{t+1}||Q_{t+1}) \le KL(P_t||Q_t), \tag{13}
$$

where the equality happens only when $P_t = Q_t$. $\qquad\square$

## B.2   Proof of Theorem 4.2

**Theorem 4.2.** *Suppose that $R(x)$ denotes a specific region of the sample $x$. Given a poisoned sample $x_0^p$, we use $R_c(x_0^p)$ and $R_c(x_0^p)$ to represent a clean region and a poisoned region of the sample, respectively. We perform the noise-based perturbation on $x_0^p$ for $\bar{t}$ steps and get the noisy samples $x_{\bar{t}}^p$, and then utilize DDPM to get a denoised sample $\widehat{x}_0^p$. When comparing different regions of the original sample $x_0^p$ with their counterparts of the detoxified sample $\widehat{x}_0^p$, we have*

$$
\frac{||R_p(\widehat{x}_0^p) - R_p(x_0^p)||^2}{n_1} > \frac{||R_c(\widehat{x}_0^p) - R_c(x_0^p)||^2}{n_2}, \tag{14}
$$

*where $n_1$ and $n_2$ represent the numbers of pixels in $R_p(x_0^p)$ and $R_c(x_0^p)$, respectively.*

*Proof:* First, we expand the inequality that we need to prove. Therefore, proving Equation 14 is equivalent to proving the following inequality:

$$
\frac{1}{n_1}\sum_{i\in[\cdot]_p}(\widehat{x}_0^p(i) - x_0^p(i))^2 > \frac{1}{n_2}\sum_{j\in[\cdot]_c}(\widehat{x}_0^p(j) - x_0^p(j))^2. \tag{15}
$$

According to Equation 5, we have

$$
x_{t-1} = \frac{1}{\sqrt{\alpha_t}}(x_t - \frac{1 - \alpha_t}{\sqrt{1 - \overline{\alpha}_t}}\theta(x_t, t)) + \sigma_t z, \tag{16}
$$

where $\sigma_t^2 = \frac{1-\overline{\alpha}_{t-1}}{1-\overline{\alpha}_t}\cdot\beta_t$ and $z \sim \mathcal{N}(0, \mathbf{I})$. $\theta$ is a pre-trained neural network. Assuming that $\theta$ is a fully trained model that can generate possible noise based on the input sample $x_t$ and the step $t$. Therefore, we have

$$
||\theta(x_t, t) - z_t|| > ||\theta(x_{t-1}, t-1) - z_{t-1}||, \tag{17}
$$

where $z_t$ is the real noise in step $t$.

Then, we analyze the pixels in the clean and poisoned regions, respectively. We denote a pixel in the clean region of the original sample as $\mathbf{x}_0$, and a pixel in the poisoned region as $\mathbf{y}_0$. Therefore, we have $\mathbf{x}_{\bar{t}}$ and $\mathbf{y}_{\bar{t}}$ to denote the pixels after our noise-based perturbation. And we have $\widehat{\mathbf{x}}_0$ and $\widehat{\mathbf{y}}_0$ to denote the pixels after our DDPM-based denoising process. Since $\theta$ has a strong ability, we can approximate the entire denoising process with the following:

$$
\begin{aligned}
\widehat{\mathbf{x}}_0 &= \frac{1}{\sqrt{\overline{\alpha}_{\bar{t}}}}(\mathbf{x}_{\bar{t}} - \sqrt{1 - \overline{\alpha}_{\bar{t}}}\cdot\mathbf{z}) \\
\widehat{\mathbf{y}}_0 &= \frac{1}{\sqrt{\overline{\alpha}_{\bar{t}}}}(\mathbf{y}_{\bar{t}} - \sqrt{1 - \overline{\alpha}_{\bar{t}}}\cdot\mathbf{z}),
\end{aligned}
\tag{18}
$$

According to Equation 2 about the relationship between $x_0$ and $x_{\bar{t}}$, we have

$$
\begin{aligned}
||\widehat{\mathbf{x}}_{\bar{t}:0} - \mathbf{x}_0|| &= ||\frac{1}{\sqrt{\overline{\alpha_{\bar{t}}}}}(\mathbf{x}_{\bar{t}} - \sqrt{1 - \overline{\alpha_{\bar{t}}}} \cdot \mathbf{z}) - \mathbf{x}_0|| \\
&= ||\mathbf{x}_0 + \frac{\sqrt{1 - \overline{\alpha_{\bar{t}}}}}{\sqrt{\overline{\alpha_{\bar{t}}}}}(\overline{\mathbf{z}}_{\bar{t}} - \mathbf{z}) - \mathbf{x}_0|| \\
&= ||\frac{\sqrt{1 - \overline{\alpha_{\bar{t}}}}}{\sqrt{\overline{\alpha_{\bar{t}}}}}(\overline{\mathbf{z}}_{\bar{t}} - \mathbf{z})||,
\end{aligned}
\tag{19}
$$

where $\overline{\mathbf{z}}_{\bar{t}}$ is the real noise and $\mathbf{z} = [z] = [\theta(x_t, t)]$. According to Equation 17, we have

$$
||\widehat{\mathbf{x}}_{t:0} - \mathbf{x}_0|| > ||\widehat{\mathbf{x}}_{t-1:0} - \mathbf{x}_0||.
\tag{20}
$$

Supposing that the training data used for the pre-trained diffusion model does not contain any knowledge about backdoor triggers, it is reasonable to assume that $\mathbf{x}_0$ belongs to the probability distribution of the training dataset of $\theta$, while $\mathbf{y}_0$ does not. Therefore, we have

$$
||\theta(\mathbf{y}_t, t) - z_t|| > ||\theta(\mathbf{x}_t, t) - z_t||.
\tag{21}
$$

According to Theorem 4.1, when the sample is fully noised, the sample is equivalent to Gaussian noise. Additionally, the sample generated by the unguided denoising process from Gaussian noise is completely independent of the original sample. Therefore, we have the equation about $\mathbf{x}$ and $\mathbf{y}$ as follows:

$$
\lim_{t \to T} ||\widehat{\mathbf{x}}_{t:0} - \mathbf{x}_0|| = \lim_{t \to T} ||\widehat{\mathbf{y}}_{t:0} - \mathbf{y}_0||.
\tag{22}
$$

According to Equation 21 and Equation 22, we finally get the following equation:

$$
0 < ||\widehat{\mathbf{x}}_{\bar{t}:0} - \mathbf{x}_0|| < ||\widehat{\mathbf{y}}_{\bar{t}:0} - \mathbf{y}_0|| < \lim_{t \to T} ||\widehat{\mathbf{x}}_{t:0} - \mathbf{x}_0|| = \lim_{t \to T} ||\widehat{\mathbf{y}}_{t:0} - \mathbf{y}_0||,
\tag{23}
$$

where $\bar{t} \in (0, T)$. Therefore, Equation 15 can be proved. Finally, we have

$$
\frac{||R_p(\widehat{x}_0^p) - R_p(x_0^p)||^2}{n_1} > \frac{||R_c(\widehat{x}_0^p) - R_c(x_0^p)||^2}{n_2}.
\tag{24}
$$

$\square$

## C  Additional Experimental Results

### C.1  Performance Comparison on More Datasets

Table 4 and Table 5 show the defense performance of our approach and three defense baselines, i.e., Sancdifi [37], BDMAE [19] and ZIP [18], against ten state-of-the-art (SOTA) attack methods on the GTSRB and Tiny-ImageNet dataset, respectively. The experimental results in the table illustrate that our approach outperforms existing SOTA methods on a variety of datasets, which shows the generalizability of our approach.

Table 4: Defense performance comparison on GTSRB, where **Bold** fonts and underlines denote the best and the second-best values.

| Defense→ | No Defense | | | Sancdifi | | | BDMAE | | | ZIP | | | SampDetox (Ours) | | |
|---|---|---|---|---|---|---|---|---|---|---|---|---|---|---|---|
| Attack ↓ | CA(%) | PA(%) | ASR(%) | CA(%) | PA(%) | ASR(%) | CA(%) | PA(%) | ASR(%) | CA(%) | PA(%) | ASR(%) | CA(%) | PA(%) | ASR(%) |
| No Attack | 98.12 | - | - | - | - | - | - | - | - | - | - | - | - | - | - |
| BadNets | 95.18 | 3.16 | 62.31 | 81.55 | 90.29 | **1.62** | 92.33 | 91.19 | 2.60 | 92.58 | 85.78 | 6.44 | **94.56** | **93.49** | 1.95 |
| SIG | 97.21 | 2.13 | 99.26 | 83.73 | 46.03 | 29.44 | 96.61 | 2.91 | 95.85 | 96.33 | 37.34 | 35.64 | **97.13** | **68.00** | 10.85 |
| LC | 97.52 | 2.33 | 99.12 | 82.48 | 53.30 | 3.86 | 93.42 | **77.36** | 1.22 | **94.29** | 74.89 | 1.99 | 94.25 | 77.03 | **1.20** |
| TrojanNN | 97.52 | 2.32 | 99.34 | 81.05 | **92.60** | 1.97 | 95.30 | 88.63 | 2.15 | 91.38 | 89.74 | 6.52 | **97.51** | 92.05 | 1.79 |
| Dynamic | 95.63 | 2.27 | 98.18 | 79.45 | 71.73 | 8.21 | 93.43 | 75.03 | 12.43 | 92.40 | 79.45 | 2.85 | **93.70** | **90.07** | 2.83 |
| Blended | 98.21 | 2.30 | 99.47 | 80.98 | 51.41 | 14.85 | 91.89 | 5.47 | 95.47 | 91.85 | 64.46 | 8.99 | **93.69** | **90.93** | 4.34 |
| LF | 97.91 | 2.33 | 96.88 | 81.86 | 51.17 | 15.98 | 92.44 | 6.57 | 80.35 | 93.08 | 87.82 | 5.18 | **94.91** | **88.64** | 2.31 |
| WaNet | 96.16 | 2.33 | 93.64 | 82.53 | 43.02 | 14.62 | 90.15 | 22.38 | 49.80 | 89.99 | 88.17 | 8.38 | **92.63** | **90.38** | 5.36 |
| ISSBA | 97.47 | 2.34 | 98.20 | 80.78 | 53.51 | 13.92 | 89.61 | 53.29 | 21.65 | 91.09 | 85.25 | **1.05** | **95.13** | **86.83** | 1.30 |
| BPP | 96.27 | 2.32 | 99.17 | 78.90 | 51.28 | 15.40 | 90.53 | 24.19 | 53.40 | 89.46 | 84.83 | 7.46 | **94.85** | **84.99** | 5.64 |

Table 5: Defense performance comparison on Tiny-ImageNet, where **Bold** fonts and underlines denote the best and the second-best values.

| Defense→ | No Defense | | | Sancdifi | | | BDMAE | | | ZIP | | | SampDetox (Ours) | | |
|---|---|---|---|---|---|---|---|---|---|---|---|---|---|---|---|
| Attack↓ | CA(%) | PA(%) | ASR(%) | CA(%) | PA(%) | ASR(%) | CA(%) | PA(%) | ASR(%) | CA(%) | PA(%) | ASR(%) | CA(%) | PA(%) | ASR(%) |
| No Attack | 56.71 | - | - | - | - | - | - | - | - | - | - | - | - | - | - |
| BadNets | 56.61 | 0.51 | 99.96 | 40.94 | 49.17 | 3.88 | 51.56 | 50.84 | 4.45 | 52.28 | 44.15 | 7.34 | **53.95** | **53.02** | **4.40** |
| SIG | 56.00 | 0.50 | 99.97 | 43.33 | 25.77 | 11.09 | 54.89 | 0.26 | 96.93 | **56.08** | 11.03 | 37.36 | 55.24 | **38.07** | **8.74** |
| LC | 56.09 | 0.51 | 99.71 | 41.89 | 12.81 | 5.03 | 53.23 | 36.63 | 3.33 | 54.00 | 33.90 | **2.19** | **54.44** | **46.67** | 2.52 |
| TrojanNN | 56.27 | 0.50 | 99.85 | 40.54 | 51.92 | 2.32 | **54.90** | 47.17 | 3.04 | 51.33 | 49.21 | 6.97 | 54.20 | **52.10** | **2.63** |
| Dynamic | 55.65 | 0.49 | 99.34 | 39.14 | 29.90 | 10.52 | 53.22 | 33.93 | 13.04 | 51.96 | 39.10 | **3.81** | **54.18** | **50.23** | 4.76 |
| Blended | 56.36 | 0.52 | 99.47 | 40.85 | 10.69 | 15.07 | 51.39 | 0.81 | 97.96 | 51.48 | 22.67 | 9.52 | **53.20** | **51.04** | **5.31** |
| LF | 56.82 | 0.51 | 99.45 | 40.92 | 9.93 | 18.86 | 51.75 | 0.42 | 82.24 | 53.07 | 47.14 | 6.03 | **54.11** | **48.91** | **4.22** |
| WaNet | 55.34 | 0.50 | 99.10 | 42.47 | 11.75 | 15.81 | 49.95 | 6.93 | 52.36 | 49.93 | 46.17 | 8.24 | **51.90** | **50.40** | **6.49** |
| ISSBA | 55.60 | 0.51 | 99.48 | 40.60 | 13.35 | 16.61 | 49.49 | 13.16 | 24.00 | 50.62 | 44.27 | **1.07** | **54.28** | **47.23** | 3.11 |
| BPP | 57.01 | 0.52 | 99.73 | 38.79 | 10.14 | 17.68 | 48.90 | 4.52 | 55.86 | 49.18 | 42.94 | 8.68 | **53.93** | **45.34** | **6.48** |

## C.2 Performance on More Models.

Since our approach does not require any interaction with the model, it should have defensive effects on different models. Table 6 and Table 7 show the defense performance of our approach on the VGG-19 and ResNet34 models on CIFAR-10, GTSRB, and Tiny-ImageNet datasets, respectively. The experimental results show that our approach has satisfactory defense effects on VGG-19 and ResNet34, which demonstrates that our approach is model-independent.

Table 6: Defense performance of our approach on VGG-19 model.

| Dataset→ | CIFAR-10 | | | GTSRB | | | Tiny-ImageNet | | |
|---|---|---|---|---|---|---|---|---|---|
| Attack↓ | CA(%) | PA(%) | ASR(%) | CA(%) | PA(%) | ASR(%) | CA(%) | PA(%) | ASR(%) |
| No Attack | 91.02 | - | - | 94.93 | - | - | 45.54 | - | - |
| BadNets | 86.82 | 86.35 | 2.42 | 91.15 | 89.98 | 2.08 | 42.12 | 39.43 | 4.31 |
| SIG | 81.62 | 61.80 | 11.39 | 94.19 | 64.33 | 10.72 | 43.51 | 24.35 | 9.01 |
| LC | 78.68 | 70.65 | 1.57 | 90.86 | 73.12 | 1.47 | 42.61 | 32.80 | 2.45 |
| TrojanNN | 90.67 | 86.19 | 1.73 | 94.42 | 87.60 | 1.42 | 42.17 | 38.88 | 2.61 |
| Dynamic | 86.06 | 85.60 | 1.73 | 90.47 | 85.68 | 3.22 | 42.41 | 36.93 | 4.44 |
| Blended | 87.91 | 83.22 | 1.57 | 90.29 | 87.05 | 4.11 | 41.34 | 37.16 | 5.01 |
| LF | 87.82 | 84.36 | 3.50 | 92.41 | 84.30 | 2.50 | 42.22 | 35.11 | 3.97 |
| WaNet | 86.40 | 85.76 | 5.45 | 90.10 | 86.80 | 5.53 | 39.88 | 36.43 | 6.47 |
| ISSBA | 88.58 | 83.39 | 2.07 | 92.37 | 83.21 | 1.23 | 42.33 | 34.18 | 2.86 |
| BPP | 88.31 | 81.59 | 6.63 | 91.38 | 80.88 | 6.01 | 41.79 | 31.76 | 6.40 |

Table 7: Defense performance of our approach on ResNet34 model.

| Dataset→ | CIFAR-10 | | | GTSRB | | | Tiny-ImageNet | | |
|---|---|---|---|---|---|---|---|---|---|
| Attack↓ | CA(%) | PA(%) | ASR(%) | CA(%) | PA(%) | ASR(%) | CA(%) | PA(%) | ASR(%) |
| No Attack | 85.34 | - | - | 97.62 | - | - | 59.96 | - | - |
| BadNets | 84.27 | 83.11 | 2.67 | 93.44 | 90.44 | 1.48 | 56.77 | 54.72 | 4.58 |
| SIG | 78.66 | 59.10 | 10.94 | 96.02 | 66.15 | 10.16 | 59.31 | 39.78 | 8.04 |
| LC | 76.22 | 67.47 | 1.56 | 92.03 | 73.94 | 1.03 | 57.39 | 48.21 | 2.45 |
| TrojanNN | 88.40 | 83.17 | 2.03 | 96.67 | 88.42 | 2.20 | 58.05 | 53.88 | 2.96 |
| Dynamic | 83.46 | 82.25 | 1.99 | 91.90 | 85.81 | 3.81 | 56.74 | 53.62 | 6.06 |
| Blended | 84.97 | 79.47 | 1.66 | 91.78 | 88.58 | 3.96 | 55.52 | 52.78 | 6.55 |
| LF | 85.37 | 81.03 | 3.86 | 94.67 | 84.47 | 3.00 | 56.83 | 51.17 | 2.76 |
| WaNet | 84.26 | 83.33 | 5.94 | 92.66 | 88.19 | 5.63 | 54.54 | 52.84 | 8.26 |
| ISSBA | 85.73 | 80.90 | 1.94 | 95.09 | 83.60 | 1.06 | 58.28 | 49.53 | 3.79 |
| BPP | 85.65 | 79.19 | 6.40 | 93.86 | 81.13 | 5.19 | 55.96 | 48.35 | 4.72 |

## C.3 Comparison with Sample Transformation Methods

In this section, we compare our approach with existing sample transformation methods on CIFAR-10, including Noising, ShrinkPad, and Blur [49]. **Noising** attempts to destroy the backdoor trigger structure by adding a large amount of noise to the sample without considering the inference accuracy. **ShrinkPad** shrinks the samples and then randomly pads samples back to their original size. **Blur** utilizes Gaussian kernel to pool the samples to destroy backdoor trigger patterns. The experimental results are shown in Table 8.

From the table, we can find that transforming the samples can break some of the backdoor triggers. However, the transformation will cause great damage to the quality of the samples, resulting in a sig-

Table 8: Defense performance comparison with sample transformation methods.

| Defense→ | No Defense | | | Noising | | | ShrinkPad | | | Blur | | | SampDetox (Ours) | | |
|---|---|---|---|---|---|---|---|---|---|---|---|---|---|---|---|
| Attack ↓ | CA(%) | PA(%) | ASR(%) | CA(%) | PA(%) | ASR(%) | CA(%) | PA(%) | ASR(%) | CA(%) | PA(%) | ASR(%) | CA(%) | PA(%) | ASR(%) |
| No Attack | 93.84 | - | - | - | - | - | - | - | - | - | - | - | - | - | - |
| BadNets | 92.00 | 10.18 | 99.97 | 68.43 | 61.79 | 5.42 | 71.63 | 66.06 | 8.42 | 64.02 | 49.44 | 18.15 | **89.57** | **90.15** | **2.11** |
| SIG | 84.94 | 9.78 | 98.50 | 60.43 | 13.97 | 91.34 | 64.19 | 59.75 | 8.35 | 55.74 | 48.50 | **6.69** | **83.71** | **65.06** | 11.03 |
| LC | 84.34 | 10.26 | 99.06 | 61.26 | 55.91 | 4.57 | 63.71 | 58.07 | 6.42 | 56.10 | 50.02 | 12.52 | **80.72** | **74.36** | **1.55** |
| TrojanNN | 93.20 | 11.07 | 99.03 | 69.20 | 59.40 | 12.16 | 73.40 | 69.17 | 4.33 | 64.67 | 50.90 | 14.30 | **92.78** | **89.95** | **1.86** |
| Dynamic | 91.09 | 10.02 | 98.19 | 67.45 | 59.94 | 1.67 | 72.76 | 64.31 | 1.67 | 63.80 | 55.76 | 2.34 | **88.52** | **88.62** | **1.45** |
| Blended | 93.85 | 10.93 | 99.51 | 71.14 | 58.87 | 25.78 | 72.93 | 68.52 | 2.54 | 68.61 | 61.76 | 7.69 | **90.23** | **86.65** | **1.96** |
| LF | 93.63 | 11.13 | 99.48 | 73.43 | 66.72 | 4.31 | 73.67 | 69.35 | 5.33 | 66.33 | 60.98 | **2.37** | **90.01** | **87.40** | 3.02 |
| WaNet | 91.43 | 10.27 | 91.05 | 68.51 | 62.95 | 5.53 | 72.52 | 65.23 | 7.52 | 65.35 | 57.25 | 5.67 | **89.34** | **88.92** | **5.59** |
| ISSBA | 93.57 | 11.38 | 95.96 | 68.96 | 62.88 | 7.82 | 73.22 | 65.40 | 1.67 | 65.36 | 55.55 | 2.02 | **90.74** | **86.51** | **1.60** |
| BPP | 91.38 | 9.46 | 98.40 | 67.36 | 51.82 | 10.65 | 72.25 | 55.43 | 7.14 | 65.86 | 56.90 | 8.34 | **90.59** | **84.83** | **6.15** |

nificant reduction in Clean sample Accuracy (CA) and Poisoned sample Accuracy (PA). Specifically, though Blur has lower ASR than SampDetox when defending against the SIG and LF attacks, our approach has better CA and PA. In addition, our methods have the best defensive performance. The experimental results show that our approach can successfully defend against various attacks while preserving the original semantic information of samples to the greatest extent.

## C.4  Performance using Different Diffusion Models

In this section, we deploy our approach using different pre-trained diffusion models to defend against backdoor attacks on CIFAR-10 and compare their defense performance. Specifically, we choose three diffusion models with different parameters: 1) pre-trained on CIFAR-10, which is the same setup as our main experiments, 2) pre-trained on ImageNet, and 3) pre-trained on ImageNet and fine-turned on CIFAR-10. Notably, we used 5% clean samples in the test dataset of CIFAR-10 to fine-tune the pre-trained model, and we excluded these data during the experiments. The experimental results are shown in Table 9.

From the table, we can observe that SampDetox with different diffusion models has satisfied defensive performance on Clean sample Accuracy (CA) and Attack Success Rate (ASR). For CA, since clean samples do not have any trigger patterns, only a little noise will be added to them in Stage 2 of our approach. Therefore, all these three diffusion models can accurately denoise samples. For ASR, the results are generally not related to the denoising process. For Poisoned sample Accuracy (PA), when defending against visible attacks (i.e., BadNets, SIG, LC, TrojanNN, and Dynamic), the model pre-trained on ImageNet has much lower accuracy, meaning that the model pre-trained on outside distribution datasets can not accurately denoise samples, especially when much noise is added. The fine-tuning model achieves satisfactory defensive performance against different attacks. Therefore, when we need to detoxify samples without pre-trained models, we can fine-tune a diffusion model to deploy our defense approach.

Table 9: Defense performance of our approach using different diffusion models.

| Diffusion→ | Pre-trained (CIFAR-10) | | | Pre-trained (ImageNet) | | | Fine-tuning (CIFAR-10) | | |
|---|---|---|---|---|---|---|---|---|---|
| Attack ↓ | CA% | PA% | ASR% | CA% | PA% | ASR% | CA% | PA% | ASR% |
| No Attack | 93.84 | - | - | 93.39 | - | - | 93.81 | - | - |
| BadNets | 89.57 | 90.15 | 2.11 | 89.23 | 85.02 | 2.26 | 89.59 | 90.77 | 1.98 |
| SIG | 83.71 | 65.06 | 11.03 | 83.27 | 56.04 | 11.05 | 83.30 | 64.68 | 10.99 |
| LC | 80.72 | 74.36 | 1.55 | 80.05 | 64.83 | 1.80 | 80.19 | 73.62 | 1.87 |
| TrojanNN | 92.78 | 89.95 | 1.86 | 92.17 | 82.12 | 1.69 | 92.66 | 90.21 | 1.58 |
| Dynamic | 88.52 | 88.62 | 1.45 | 87.97 | 80.44 | 1.80 | 88.46 | 88.93 | 1.01 |
| Blended | 90.23 | 86.65 | 1.96 | 89.42 | 83.64 | 1.54 | 89.81 | 85.92 | 1.64 |
| LF | 90.01 | 87.40 | 3.02 | 89.38 | 86.94 | 2.65 | 89.72 | 86.83 | 3.53 |
| WaNet | 89.34 | 88.92 | 5.59 | 88.94 | 87.88 | 5.17 | 89.43 | 88.02 | 6.16 |
| ISSBA | 90.74 | 86.51 | 1.6 | 90.25 | 85.31 | 1.57 | 90.29 | 85.53 | 1.70 |
| BPP | 90.59 | 84.83 | 6.15 | 90.96 | 84.01 | 5.95 | 91.24 | 85.12 | 5.34 |

Additionally, considering the case where pre-training datasets for diffusion models are out-of-distribution from their target classification tasks, we conducted experiments to evaluate the performance of SampDetox on a subset of the MS-Celeb-1M [50] dataset. Specifically, we considered the top 100 labels with the largest number of samples and randomly selected 380 samples for each label. We split the training and tests with a ratio of 8 : 2 and adjusted the shape of the samples to

$224 * 224$. We adopted BadNets [22] and WaNet [28] as attack methods, representing a visible and an invisible backdoor attack, respectively. We considered four pre-trained diffusion models: i) a pre-trained model (i.e., Model 1) on the subset of MS-Celeb-1M, ii) a pre-trained model (i.e., Model 2) on another subset of MS-Celeb-1M, iii) a pre-trained model (i.e., Model 3) on ImageNet, and iv) Model 4, which is fine-tuned from Model 3 using a subset of MS-Celeb-1M.

Table 10 shows the experimental results. We can find that SampDetox with different diffusion models achieves satisfied defensive performance on ASR. Note that, for Model 3, the CA and PA of SampDetox decrease. This is because Model 3 is pre-trained on ImageNet rather than MS-Celeb-1M, resulting in the diffusion model failing to effectively denoise MS-Celeb-1M samples. However, we can find significant improvements in the CA and PA of Model 4 since Model 4 is fine-tuned using a subset of MS-Celeb-1M. In other words, SampDetox can defend against attacks using out-of-distribution diffusion models with proper fine-tuning.

Table 10: Defense performance of our approach using out-of-distribution diffusion models.

| Attack→ | BadNets | | | WaNet | | |
|---|---|---|---|---|---|---|
| Setting↓ | CA(%) | PA(%) | ASR(%) | CA(%) | PA(%) | ASR(%) |
| No Defense | 94.38 | 1.27 | 99.32 | 94.61 | 1.96 | 97.05 |
| Model 1 | 92.06 | 91.65 | 1.36 | 91.83 | 91.12 | 1.29 |
| Model 2 | 90.27 | 89.73 | 1.31 | 89.92 | 90.05 | 1.22 |
| Model 3 | 55.47 | 52.06 | 1.57 | 72.35 | 62.26 | 1.25 |
| Model 4 | 88.56 | 89.27 | 1.47 | 89.43 | 89.32 | 1.30 |

## D  Additional Discussions

### D.1  Performance on High-frequency Classification Task

To evaluate the defense performance of SampDetox on classification tasks that rely on benign high-frequency details, we conducted experiments on the DTD (Describable Textures Dataset) dataset [51], where we adopted a visible backdoor attack, i.e., BadNets [22], and an invisible backdoor attack, i.e., WaNet [28], respectively. Table 11 compares SampDetox with six baselines. From this table, we can find that though the CA of SampDetox is a little smaller than the CA of "No Defense", SampDetox can achieve the best defense performance on CA, PA, and ASR compared with all baselines, which shows the applicability and superiority of SampDetox.

Table 11: Defense performance of our approach on the high-frequency classification task.

| Attack→ | BadNets | | | WaNet | | |
|---|---|---|---|---|---|---|
| Defense ↓ | CA(%) | PA(%) | ASR(%) | CA(%) | PA(%) | ASR(%) |
| No Defense | 65.42 | 2.05 | 90.56 | 64.72 | 2.17 | 88.56 |
| Sancdifi | 48.98 | 51.23 | 2.84 | 49.64 | 28.06 | 15.12 |
| BDMAE | 56.25 | 56.43 | 1.86 | 55.57 | 19.17 | 47.22 |
| ZIP | 43.17 | 40.95 | 3.32 | 44.87 | 40.56 | 3.31 |
| Noising | 37.55 | 35.87 | 5.26 | 38.46 | 37.08 | 3.27 |
| ShrinkPad | 41.27 | 39.36 | 8.92 | 43.19 | 42.48 | 2.92 |
| Blur | 33.17 | 30.92 | 12.05 | 33.49 | 29.50 | 3.67 |
| SampDetox (Ours) | **57.46** | **57.12** | **1.73** | **58.33** | **58.12** | **2.21** |

Table 12: Defense performance of SampDetox using DDPM and DDIM.

| Attack | SampDetox+DDPM | | | SampDetox+DDIM | | |
|---|---|---|---|---|---|---|
| | CA(%) | PA(%) | ASR(%) | CA(%) | PA(%) | ASR(%) |
| BadNets | 89.57 | 90.15 | 2.11 | 89.49 | 90.13 | 2.12 |
| SIG | 83.71 | 65.06 | 11.03 | 83.82 | 65.13 | 10.98 |
| LC | 80.72 | 74.36 | 1.55 | 80.62 | 74.22 | 1.53 |
| TrojanNN | 92.78 | 89.95 | 1.86 | 92.83 | 89.87 | 1.69 |
| Dynamic | 88.52 | 88.62 | 1.45 | 88.52 | 88.72 | 1.42 |
| Blended | 90.23 | 86.65 | 1.96 | 90.15 | 86.54 | 2.02 |
| LF | 90.01 | 87.40 | 3.02 | 90.09 | 87.61 | 3.10 |
| WaNet | 89.34 | 88.92 | 5.59 | 89.48 | 88.82 | 5.54 |
| ISSBA | 90.74 | 86.51 | 1.60 | 90.76 | 86.65 | 1.55 |
| BPP | 90.59 | 84.83 | 6.15 | 90.42 | 84.91 | 6.17 |

## D.2 Performance using DDIM

Since DDIM [21] was proposed to accelerate the speed of DDPM without affecting the image generation quality, the defense performance of "SampDetox+DDIM" is comparable to that of "SampDetox+DDPM". To demonstrate the performance of "SampDetox+DDIM", we conducted new experiments on CIFAR-10 against ten SOTA backdoor attacks. The experimental results are shown in Table 12. From the table, we can find that the defense performance of "SampDetox+DDIM" is almost the same as that of "SampDetox+DDPM", which means that "SampDetox+DDIM" can match the defense efficacy of "SampDetox+DDPM".

## D.3 Training Sample Detoxification

Before training deep models, we can use SampDetox to conduct perturbation-based training sample detoxification, protecting them against backdoor injection. Table 13 compares the defense performance of the trained models by SampDetox. Here, the second column shows the defense performance without using SampDetox. The third and fourth columns present the defense performance of models trained by detoxified samples. Note that for the third column, we did not change the labels of detoxified samples. In other words, the poisoned samples still keep their dirty labels (denoted by *D.L.*). For the fourth column, we tried to correct the dirty labels based on the inference results of the models generated by the third column. Based on the detoxified samples and corrected clean labels (denoted by *C.L.*), the fourth column shows the defense performance of our approach. Note that for this table, we did not apply sample detoxification on test samples. From this table, we can observe that SampDetox can effectively defend against various backdoor attacks. Although "SampDetox+D.L." and "SampDetox+C.L." have similar ASR, the CA of "SampDetox+C.L." can be notably improved based on the corrected labels of poisoned samples.

Table 13: Defense performance of SampDetox in the training phase.

| Defense→ | No defense | | SampDetox+D.L. | | SampDetox+C.L. | |
|---|---|---|---|---|---|---|
| Attack ↓ | CA(%) | ASR(%) | CA(%) | ASR(%) | CA(%) | ASR(%) |
| BadNets | 91.95 | 99.97 | 88.97 | 1.46 | 92.95 | 1.41 |
| SIG | 85.03 | 98.39 | 87.33 | 1.78 | 91.53 | 1.69 |
| LC | 84.64 | 99.06 | 89.78 | 1.49 | 89.78 | 1.49 |
| TrojanNN | 93.37 | 99.99 | 89.46 | 2.01 | 93.10 | 1.84 |
| Dynamic | 91.05 | 98.12 | 88.19 | 1.64 | 90.98 | 1.89 |
| Blended | 93.41 | 99.96 | 87.67 | 1.45 | 91.59 | 1.46 |
| LF | 93.11 | 99.31 | 88.59 | 1.63 | 90.77 | 1.45 |
| WaNet | 91.81 | 90.77 | 88.18 | 1.77 | 90.85 | 1.85 |
| ISSBA | 93.09 | 96.36 | 89.79 | 1.55 | 93.31 | 1.60 |
| BPP | 91.39 | 99.18 | 89.52 | 1.47 | 93.21 | 1.38 |

## D.4 Performance against Adaptive Attacks

Assume that attackers have all the prior knowledge of our proposed approach to design an adaptive attack to bypass our defense approach. Since SampDetox is based on observations about the correlation between the visibility of triggers and the robustness of poisoned samples, the attackers can try two types of adaptive attacks as follows.

**1) Adaptive attack 1.** The attacker uses an attack with both low visibility and high robustness since attacks with high visibility and low robustness are easily defended. To guarantee low visibility, the attacker adopts an invisible backdoor attack, i.e., WaNet. To ensure high robustness, the attacker adaptively adds noises to samples during the training phase to improve the robustness of triggers.

**2) Adaptive attack 2.** The attacker does not consider visibility issues and tries to maximize robustness to bypass our defense. In this case, the attacker adopts a visible backdoor attack, i.e., BadNets, and adds robust noise during the training phase to improve robustness.

We conducted experiments to defend against the two adaptive attacks on the CIFAR-10 dataset. Table 14 shows the results of the experiments, where the first column denotes the amount of noise added during training to improve the robustness. From the table, we find that our approach can defend against adaptive attacks.

Table 14: Defense performance of SampDetox against adaptive attacks.

| Robust Noise | Attack 1 (No Defense) | | | Attack 1 (SampDetox) | | | Attack 2 (No Defense) | | | Attack 2 (SampDetox) | | |
|---|---|---|---|---|---|---|---|---|---|---|---|---|
| $\eta$ | CA(%) | PA(%) | ASR(%) | CA(%) | PA(%) | ASR(%) | CA(%) | PA(%) | ASR(%) | CA(%) | PA(%) | ASR(%) |
| 0 | 91.43 | 10.27 | 91.05 | 89.34 | 88.92 | 5.59 | 92.00 | 10.18 | 99.97 | 89.57 | 90.15 | 2.11 |
| 0.10 | 91.62 | 13.14 | 87.36 | 89.45 | 89.07 | 5.22 | 92.16 | 10.07 | 99.12 | 89.62 | 90.37 | 2.25 |
| 0.20 | 91.87 | 21.18 | 52.53 | 90.15 | 89.45 | 3.13 | 91.87 | 10.56 | 97.12 | 89.58 | 89.98 | 2.16 |
| 0.30 | 92.26 | 89.61 | 1.36 | 90.83 | 90.72 | 1.13 | 91.62 | 12.32 | 82.12 | 90.02 | 90.09 | 2.33 |

## D.5 Limitation

Although our approach aims to defend against backdoor attacks with any kind of trigger, there are still certain triggers that SampDetox cannot remove. Physical attack [52, 53] is a backdoor attack with visible but natural triggers, usually physical objects. Since our approach identifies the positions of trigger patterns based on Theorem 4.2 by calculating the similarity between the denoised and original samples, the effectiveness of our approach depends on the pre-trained diffusion model. When the triggers are visible, are part of the sample classification characteristics, and belong to the knowledge contained in the pre-trained diffusion model, our defense approach cannot destroy the trigger patterns and defend against these attacks.

Table 15: Defense performance of SampDetox against semantics attacks.

| Attack | CIFAR-10 | | | GTSRB | | | Tiny-ImageNet | | |
|---|---|---|---|---|---|---|---|---|---|
| Defense | CA(%) | PA(%) | ASR(%) | CA(%) | PA(%) | ASR(%) | CA(%) | PA(%) | ASR(%) |
| Refool | 87.46 | 88.03 | 1.72 | 93.79 | 91.43 | 2.16 | 54.52 | 51.11 | 4.10 |
| DSFT | 86.71 | 86.55 | 1.96 | 90.45 | 89.20 | 2.05 | 50.68 | 49.85 | 4.15 |

To investigate the limitation of SampDetox, we conducted experiments with three semantic backdoor attacks, i.e., Refool [54], DSFT [55] and CBA [56]. Table 15 presents the defense performance of SampDetox against Refool and DSFT. From the table, we can find that SampDetox can effectively defend against such backdoor attacks. This is because the semantic features adopted by Refool and DSFT are neither part of the original classified features nor are their distributions the same as the distribution of the training dataset of the pre-trained diffusion model. Therefore, the triggers based on such semantic features will not be restored by the denoising phase of SampDetox.

Additionally, to explain why SampDetox can defend against these two attacks, we conducted experiments to calculate the robustness of their poisoned samples, respectively. For each attack, we randomly selected 50 poisoned samples on CIFAR-10 and calculated the robustness value of each sample following our robustness definition in Section 3.1. For Refool, the robustness of its poisoned samples has a mean value of 0.0759 and a standard deviation of 0.011. For DFST, the robustness of its poisoned samples has a mean value of 0.137 and a standard deviation of 0.019. From the experimental results, we can find that the robustness of the poisoned samples is low (ones with robustness < 0.18, according to Figure 2). Therefore, when SampDetox adds enough noise to these samples, even if trigger patterns are partially restored by diffusion models, backdoors will not be successfully triggered.

However, SampDetox fails to defend against CBA. Since CBA concatenates two natural images as backdoor triggers, it is considered the backdoor attack type that SampDetox fails to defend. The experimental results on CIFAR-10 are as follows: Without defense, CA, PA, and ASR were 87.52%, 10.16%, and 90.17%, respectively. However, when using SampDetox, CA, PA, and ASR were 86.81%, 38.45%, and 37.28%, respectively. The experimental results show that SampDetox has a certain effect in defending against CBA but fails to minimize ASR.

# E    More Examples of Poisoned Samples

Figure 6 shows examples of the poisoned samples with their visibility $v$ and robustness $\eta_r$.

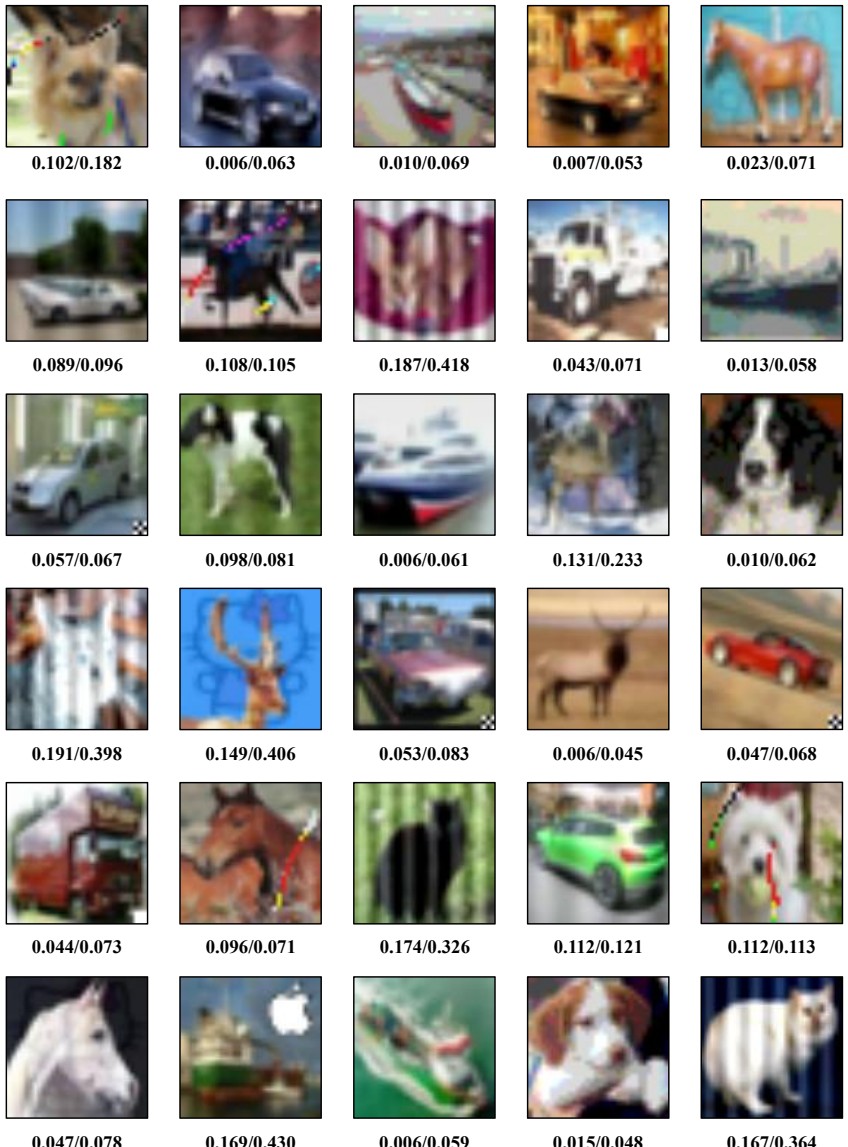

Figure 6: Examples of poisoned samples and their $v/\eta_r$.

