# OpenReview forum: "SampDetox: Black-box Backdoor Defense via Perturbation-based Sample Detoxification"
_NeurIPS.cc/2024/Conference — NeurIPS 2024 poster_

### Official Review · Reviewer_i7qR · 2024-06-26

**Soundness:** 3
**Presentation:** 3
**Contribution:** 3
**Rating:** 6
**Confidence:** 4

**Summary:**

The paper addresses the issue of backdoor threats in third-party Machine Learning as a Service (MLaaS) applications. The authors propose a novel two-stage method called SampDetox to defend against both visible and invisible backdoor attacks without requiring access to the model's parameters or training data. The proposed method uses diffusion models to introduce noise into samples to destroy backdoors and recover clean samples. It combines lightweight and high noise strategically to handle backdoors with varying visibility and robustness. The paper provides mathematical proofs to support the effectiveness of SampDetox in eliminating backdoor triggers and restoring original samples. Comprehensive experiments demonstrate SampDetox's effectiveness against various state-of-the-art backdoor attacks. The method is evaluated using different datasets and models, showing its superiority over existing defense methods in terms of clean sample accuracy, poisoned sample accuracy, and attack success rate.

**Strengths:**

The strengths can be summarized as follows:

1. **Innovative Defense Mechanism**: SampDetox introduces a novel two-stage approach to address backdoor attacks in a black-box context, which is a significant contribution to the field of machine learning security.

2. **Theoretical Foundation**: The paper provides a solid theoretical basis with mathematical proofs that support the effectiveness of the proposed method in eliminating backdoor triggers.

3. **Comprehensive Evaluation**: The method is extensively tested against state-of-the-art backdoor attacks using multiple datasets and models, demonstrating its generalizability and reliability.

4. **Thorough Analysis**: The paper includes ablation studies that dissect the impact of different stages and parameters of the SampDetox method, providing insights into its workings and optimization.

**Weaknesses:**

**Summary of Key Weaknesses:**

1. **Dependency on Pre-trained Models**:
   - *SampDetox's* effectiveness is contingent upon the quality of pre-trained diffusion models utilized for the denoising process. This poses a limitation, particularly when these models are not finely tuned for the specific data in question. While the authors have successfully demonstrated *SampDetox* using an ImageNet-trained model to counter attacks on CIFAR10, the real-world applicability may be limited due to the difficulty of accessing a pre-trained model that matches the unique dataset of the application.

2. **Trigger Recovery Assumption**:
   - The paper presupposes that clean regions in poisoned samples can be easily restored, whereas poisoned regions, especially with visible triggers, cannot (as stated on Line 201). This assumption may not hold against sophisticated attacks like physical attacks where triggers are inherent and visible parts of the sample. Furthermore, this assumption could be invalidated if the image features an object that the pre-trained diffusion model is incapable of accurately recovering.

3. **Omission of Experimental Details**:
   - The paper falls short in providing certain critical experimental details, particularly regarding the baseline defense methods (such as the diffusion model configuration for ZIP) and the proposed method itself ($\bar{t}_1$ and $\bar{t}_2$ for different dataset).

**Questions:**

The Questions can be summarized as follows:

1. **Question on Figure 6**: The authors compare the additional time overhead for various methods in Figure 6, noting that ZIP's overhead exceeds SampDetox's by more than 20 times. Given that ZIP utilizes DDPM once for image purification, whereas SampDetox employs it twice, this discrepancy seems disproportionate, especially if both are configured for the same denoising step count.  If ZIP and SampDetox use different configurations of DDPM, such comparison in overhead and defense performance may not be fair in some sense, as the configuration of DDPM plays an essential role in the recovery performance. Clarification on this point is requested.

2. **Lack results for SampDetox with DDIM**: In section 5.4, the authors assert that integrating DDIM with SampDetox yields inference times comparable to unguarded methods. However, specific performance metrics for SampDetox, when augmented with DDIM, are absent. To substantiate this claim, it is imperative to demonstrate that SampDetox+DDIM can match the defense efficacy of SampDetox+DDPM.

3. **Question on Table 5**: Table 5 presents ablation study outcomes for varying parameter selections. It remains ambiguous whether the data pertain to a single specific attack scenario or represent an average across multiple attack types.

4. **The stability of parameters across different datasets**: Does the suggest parameter for $\hat{t}_1=20$ and  $\hat{t}_2=120$ works consistently across different datasets?

I will raise my score if the author can address my concerns.

**Limitations:**

The authors have acknowledged certain limitations but have not sufficiently addressed the challenges associated with accessing pre-trained diffusion models. This is a critical issue that requires further exploration. A primary concern is the need for additional experiments to validate the approach using diffusion models trained on varied datasets. For instance, testing a model trained on ImageNet to defend against attacks on the CelebA dataset would provide valuable insights into the model's adaptability and robustness (Just as a suggestion/example).

---

> ### Author Rebuttal · Authors · 2024-08-06
>
> **AW1:**
> Thanks for your suggestion. We conducted new experiments to evaluate the performance of SampDetox considering the case where pre-training datasets for diffusion models are out-of-distribution from their target classification tasks. The detailed experimental setups are as follows:
> * We conducted the training on a subset of the MS-Celeb-1M dataset. Specifically, we considered the top 100 labels with the largest number of samples and randomly selected 380 samples for each label. We split the training and tests with a ratio of 8:2 and adjusted the shape of the samples to 224*224.
> * We adopted BadNets and WaNet as attack methods, representing a visible and an invisible backdoor attack, respectively.
> * We considered four pre-trained diffusion models: i) a pre-trained model (i.e., **Model 1**) on one subset of MS-Celeb-1M, ii) a pre-trained model (i.e., **Model 2**) on another subset of MS-Celeb-1M, iii) a pre-trained model (i.e., **Model 3**) on ImageNet, and iv) **Model 4**, which is fine-tuned from Model 3 using a subset of MS-Celeb-1M.
>
> The following table shows the experimental results. We can find that SampDetox with different diffusion models achieves satisfied defensive performance on ASR. Note that, for Model 3, the CA and PA of SampDetox decrease. This is because Model 3 is pre-trained on ImageNet rather than MS-Celeb-1M, resulting in the diffusion model failing to effectively denoise MS-Celeb-1M samples. However, we can find significant improvements in the CA and PA of Model 4 since Model 4 is fine-tuned using a subset of MS-Celeb-1M. In other words, SampDetox can defend against attacks using out-of-distribution diffusion models with proper fine-tuning.​
>
> |Diffusion Models|BadNets|WaNet|
> |:-:|:-:|:-:|
> ||CA(%)/PA(%)/ASR(%)|CA(%)/PA(%)/ASR(%)|
> |No Defense|94.38/1.27/99.32|94.61/1.96/97.05|
> |Model1|92.06/91.65/1.36|91.83/91.12/1.29|
> |Model2|90.27/89.73/1.31|89.92/90.05/1.22|
> |Model3|55.47/52.06/1.57|72.35/62.26/1.25|
> |Model4|88.56/89.27/1.47|89.43/89.32/1.30|
>
> **AW2:**
> Sorry for your confusion. In Appendix D.3 (Limitations of SampDetox), we have discussed a type of backdoor attack that SampDetox cannot defend against. Specifically, when dealing with triggers derived from semantic features that are part of the original classified features and whose distribution is the same as the distribution of the training dataset of the pre-trained diffusion model, these triggers will be restored by the denoising phase of SampDetox. In this case, the triggers will not be destroyed by SampDetox, resulting in a failure of defense.
>
> **AW3:**
> Thanks for your suggestion. The details are as follows.
> * SampDetox (Ours). As mentioned in Appendix A.2, we utilized the improved diffusion model provided by [17], whose diffusion process consists of 1000 steps. We use the same settings $\overline{t}_1=20$ and $\overline{t}_2=120$ throughout the paper.
> * Sancdifi followed the settings in Section “Numerical Experiments” of its paper [33]. Sancdifi computed RISE maps using 2000 random binary masks and set the saliency threshold to 0.95. For a fair comparison, Sancdifi used the same diffusion models as ours.
> * BDMAE followed the settings in Section “Method Configurations” of its paper [15], which used masked autoencoders pre-trained on ImageNet with 24 encoder layers.
> * ZIP followed the settings in Section “Purification Implementation” of its paper [14], which set its hyperparameter $\lambda$ to 2. For a fair comparison, ZIP used the same diffusion models as ours.
>
> We will detail the above experimental settings in the final version.
>
> **AQ1:**
> Sorry for your confusion.
> It should be clarified that we used the same DDPM configuration for ZIP and SampDetox in our experiments. SampDetox is much faster than ZIP because ZIP relies on the whole diffusion process of DDPM, while SampDetox only needs partial steps of the whole diffusion process. Specifically, ZIP utilizes DDPM to generate samples from completely random noise, requiring the whole diffusion process with 1000 denoising steps. However, since we set $\overline{t}_1=20$, SampDetox requires only 20 denoising steps in Stage 1. For Stage 2, the number of denoising steps varies from pixel to pixel, but it will not exceed 120 steps at most due to $\overline{t}_2=120$. In this way, though SampDetox employs DDPM twice, its overhead is still much less than that of ZIP.
>
> **AQ2:**
> Thanks for your suggestion. Since DDIM was proposed to accelerate the speed of DDPM without affecting the image generation quality, the defense performance of **SampDetox+DDIM** is comparable to that of **SampDetox+DDPM**. To demonstrate the performance of **SampDetox+DDIM**, we conducted new experiments on CIFAR-10 against ten SOTA backdoor attacks. From the following table, we can find that the defense performance of **SampDetox+DDIM** is almost the same as that of **SampDetox+DDPM**, which means that **SampDetox+DDIM** can match the defense efficacy of **SampDetox+DDPM**.
>
> |Attack Methods|SampDetox+DDPM|SampDetox+DDIM|
> |:-:|:-:|:-:|
> ||CA(%)/PA(%)/ASR(%)|CA(%)/PA(%)/ASR(%)|
> |BadNets|89.57/90.15/2.11|89.49/90.13/2.12|
> |SIG|83.71/65.06/11.03|83.82/65.13/10.98|
> |LC|80.72/74.36/1.55|80.62/74.22/1.53|
> |TrojanNN|92.78/89.95/1.86|92.83/89.87/1.69|
> |Dynamic|88.52/88.62/1.45|88.52/88.72/1.42|
> |Blended|90.23/86.65/1.96|90.15/86.54/2.02|
> |LF|90.01/87.40/3.02|90.09/87.61/3.10|
> |WaNet|89.34/88.92/5.59|89.48/88.82/5.54|
> |ISSBA|90.74/86.51/1.60|90.76/86.65/1.55|
> |BPP|90.59/84.83/6.15|90.42/84.91/6.17|
>
> **AQ3:**
> Sorry for the confusion. The results in Table 5 represent the average results across multiple attack types. We will add this information in the final version.
>
> **AQ4:**
> Sorry for the confusion.
> The setting of $\overline{t}_1=20$ and $\overline{t}_2= 120$ remains consistent across varying datasets and models. In other words, we use the same fixed settings throughout the paper.
>
> **AL1:**
> Thanks for your suggestion. Please refer to **AW1** for our answer.

---

> > ### Comment · Reviewer_i7qR · 2024-08-07
> > **Update**
> >
> > The authors' rebuttal has addressed most of my concerns, so I have increased my score.
> >
> > I encourage the authors to release their code to the community in the future, which would be a valuable contribution to this field.

---

> > > ### Author Response · Authors · 2024-08-09
> > >
> > > Dear Reviewer i7qR,
> > >
> > > Thank you once again for taking the time to review our work. We sincerely appreciate your thorough feedback and valuable insights. We commit to releasing our code to the community.
> > >
> > > Best Regards,
> > >
> > > The authors of 8771

---

### Official Review · Reviewer_2sKD · 2024-07-10

**Soundness:** 3
**Presentation:** 3
**Contribution:** 3
**Rating:** 6
**Confidence:** 4

**Summary:**

The paper addresses the significant issue of backdoor defense in a black-box setting during the inference phase of MLaaS applications. It proposes a novel technique called SampDetox, where the method uses diffusion models for detoxification by injecting noise to disrupt backdoors and retrieve clean instances.

**Strengths:**

1. The work underlines an essential research value and a practical scenario by studying black-box backdoor defense during the inference phase. The proposed two-tier defense method shows promise, backed by theoretical underpinnings, and demonstrates efficacy with relatively minimal overhead.

2. An extensive set of experiments validate the effectiveness of the proposed method, surpassing existing defenses. Notably, the method can lower the Attack Success Rate of backdoor models and maintain a high PA on backdoor samples.

**Weaknesses:**

1. The research appears to make rather robust assumptions for diffusion models. The authors assume that the clean regions of the poisoned samples can be easily restored, while the poisoned regions, especially highly visible triggers, may fail to do so (as empathized in Line 201-202). This could be because the diffusion model learns only from the clean samples and fails to learn backdoor information. While the paper conducts supplemental experiments demonstrating strong defense performance even under other diffusion models (as shown in **Table 8**), the generalizability from ImageNet to CIFAR-10 seems less convincing. If the diffusion model is backdoored or the pre-trained diffusion model is trained on out-of-distribution samples, it could potentially dampen the final defense performance.

2. Furthermore, the method is less likely to be effective if physical attacks are deployed, where natural objects are used as backdoor triggers, as diffusion could learn features of these natural triggers.

**Questions:**

Even though there is a theoretical substantiation for the working defense, I wonder about the universality of this theory in real-world applications. Specifically, in varying datasets and models, would the hyper-parameters \bar{t}_1 and \bar{t}_2 require fine-tuning for different noise levels? While **Figure 5** provides an ablation study for different \bar{t}_1 and \bar{t}_2, it does not display the specific setting. It would be intriguing to know whether different \bar{t}_1 and \bar{t}_2 were utilized for the results in **Tables 1, 3, 4, 5, and 6**.

**Limitations:**

Although the paper acknowledges its limitations, a major overlooked limitation lies in the pre-trained diffusion model. Would the detoxifying effect still be substantial if the diffusion model used is trained on out-of-distribution samples compared to the target dataset? This question presents an area of potential further exploration to validate the robustness of the proposed SampDetox method.

---

> ### Author Rebuttal · Authors · 2024-08-06
>
> **AW1:**
> Thanks for your suggestion.
> First, it should be clarified that the defender completely controls the design, training, and utilization of diffusion models. Therefore, it is almost impossible for attackers to inject backdoors into diffusion models in practice.
> In addition, even if the training data for the diffusion model is poisoned, backdoors can not be injected into it because attackers need to modify the training process of the diffusion models to inject backdoors [1].
> Moreover, even if the diffusion model is backdoored, specific trigger patterns must be added to the input Gaussian noise of the diffusion model to trigger the backdoor.  However, when the input of SampDetox is Gaussian noise, due to the small number of denoising steps of SampDetox (i.e., $\overline{t}_1=20, \overline{t}_2=120$), it is extremely hard for a backdoored diffusion model to generate its target image.
>
> In conclusion, backdoored diffusion models do not compromise the defense performance of SampDetox. Thanks again for your suggestion. We will add the discussion in the final version.
>
> **Reference:**
> [1] Chou, Sheng-Yen, et al. How to Backdoor Diffusion Models? CVPR’23
>
> We conducted new experiments to evaluate the performance of SampDetox considering the case where pre-training datasets for diffusion models are out-of-distribution from their target classification tasks. The detailed experimental setups are as follows:
> * We conducted the training on a subset of the MS-Celeb-1M dataset. Specifically, we considered the top 100 labels with the largest number of samples and randomly selected 380 samples for each label. We split the training and tests with a ratio of 8:2 and adjusted the shape of the samples to 224*224.
> * We adopted BadNets and WaNet as attack methods, representing a visible and an invisible backdoor attack, respectively.
> * We considered four pre-trained diffusion models: i) a pre-trained model (i.e., **Model 1**) on one subset of MS-Celeb-1M, ii) a pre-trained model (i.e., **Model 2**) on another subset of MS-Celeb-1M, iii) a pre-trained model (i.e., **Model 3**) on ImageNet, and iv) **Model 4**, which is fine-tuned from Model 3 using a subset of MS-Celeb-1M.
>
> The following table shows the experimental results. We can find that SampDetox with different diffusion models achieves satisfied defensive performance on ASR. Note that, for Model 3, the CA and PA of SampDetox decrease. This is because Model 3 is pre-trained on ImageNet rather than MS-Celeb-1M, resulting in the diffusion model failing to effectively denoise MS-Celeb-1M samples. However, we can find significant improvements in the CA and PA of Model 4 since Model 4 is fine-tuned using a subset of MS-Celeb-1M. In other words, SampDetox can defend against attacks using out-of-distribution diffusion models with proper fine-tuning.​
> ​
>
> | Diffusion Models | &emsp;&emsp;BadNets | &emsp;&emsp;WaNet |
> | :----------------: | :------------------------: | :------------------------: |
> | | CA(%) / PA(%) / ASR(%) | CA(%) / PA(%) / ASR(%) |
> | No Defense |  94.38 / 1.27 / 99.32 | 94.61 / 1.96 / 97.05 |
> | Model 1 | 92.06 / 91.65 / 1.36 | 91.83 / 91.12 / 1.29 |
> | Model 2 | 90.27 / 89.73 / 1.31 | 89.92 / 90.05 / 1.22 |
> | Model 3 | 55.47 / 52.06 / 1.57 | 72.35 / 62.26 / 1.25 |
> | Model 4 | 88.56 / 89.27 / 1.47 | 89.43 / 89.32 / 1.30 |
>
> **AW2:**
> Sorry for your confusion.
> In Appendix D.3 (Limitations of SampDetox), we have discussed a type of backdoor attack that SampDetox cannot defend against. Specifically, when dealing with triggers derived from semantic features that are part of the original classified features and whose distribution is the same as the distribution of the training dataset of the pre-trained diffusion model, these triggers will be restored by the denoising phase of SampDetox. In this case, the triggers will not be destroyed by SampDetox, resulting in a failure of defense.
>
> **AQ1:**
> Sorry for the confusion. Based on the ablation study results shown in Figure 5, we suggest to set
> $\overline{t}_1$ and $\overline{t}_2 $ to 20 and 120, respectively. Please see lines 383-384 for more details. Please note that we use the same settings, i.e., $\overline{t}_1=20$ and $\overline{t}_2 = 120$ throughout the paper. In other words, the experimental results in Tables 1, 3, 4, 5, and 6 are obtained utilizing the same $\overline{t}_1$ and $\overline{t}_2$.
>
> **AL1:** Thanks for your suggestion. Please refer to **AW1** for our answer.

---

> > ### Comment · Reviewer_2sKD · 2024-08-10
> >
> > The author's responses to W1 and W2 addressed my concerns. Thus I raised the rating. I recommend incorporating these discussions into the final version, as this would make the paper more comprehensive and persuasive.

---

> > > ### Author Response · Authors · 2024-08-10
> > >
> > > Dear Reviewer 2sKD,
> > >
> > > Thank you once again for taking the time to review our work. We sincerely appreciate your thorough feedback and valuable insights. The discussions will be included in the final version of the paper.
> > >
> > >
> > > Best Regards,
> > >
> > > The authors of 8771

---

### Official Review · Reviewer_gGGP · 2024-07-11

**Soundness:** 3
**Presentation:** 3
**Contribution:** 3
**Rating:** 6
**Confidence:** 3

**Summary:**

This paper proposes a two-stage black-box method to defend against the backdoor in DNN. The first stage mitigates the backdoor triggers with low  visibility and the second stage mitigates the trigger with higher robustness.

**Strengths:**

1. The two-stage design is novel and effective.
2. The evaluation is comprehensive and convincing.
3. The paper is easy to read and follow.

**Weaknesses:**

1. One major concern about this work is that I think this work may not be effective against high-level semantic triggers. For example [1] adds the digital and physical triggers to the image, and make sure they can be preserved after gaussian blur or print and recapture. This work may not be effective against such triggers. Similar works can be found in [2,3]. However, I found that the authors have already mentioned  and acknowledged this in the paper. So I think this is not a big issue.
2. The use of diffusion model is not well introduced. It may not be easy for the readers to understand this part if not familiar with the diffusion model. Also, in appendix A.2, the diffusion model from previous work is used, but the authors did not provide the details of the diffusion model. I think it would be better if the authors can provide more details about the diffusion model. Also, would it have any impact if this used diffusion model is replaced by other pre-trained models?


[1] Versatile Backdoor Attack with Visible, Semantic, Sample-specific and Compatible Triggers

[2] Backdoor Attacks Against Deep Learning Systems in the Physical World

[3] Dangerous Cloak: Natural Backdoor Attacks on Object Detector in Real Physical World

**Questions:**

Would using another pre-trained model have an influence on your work?

**Limitations:**

The authors discussed about the main concern I have and acknowledged the limitation for semantic backdoor triggers.

---

> ### Author Rebuttal · Authors · 2024-08-06
>
> **AW1:**
> Indeed. In Appendix D.3 (Limitations of SampDetox), we have discussed a type of backdoor attack that SampDetox cannot defend against. Specifically, when dealing with triggers derived from semantic features that are part of the original classified features and whose distribution is the same as the distribution of the training dataset of the pre-trained diffusion model, these triggers will be restored by the denoising phase of SampDetox. In this case, the triggers will not be destroyed by SampDetox, resulting in a failure of defense.
>
> **AW2:**
> Thanks for your suggestion. We will detail the use of diffusion models in the final version.
>
> In Appendix A.2, we have mentioned that the diffusion models used the same settings as the ones presented in [17]. In this experiment, the number of total diffusion steps was set to 1000 for all the datasets in the experiments, the noise_schedule was set as ``cosine’’, and the learning rate was set to 1e-4. We will provide such details in the final version.
>
> In Appendix C.4, we have evaluated the defense performance of SampDetox using different diffusion models. Specifically, for CIFAR-10 classification tasks, we considered three different pre-trained diffusion models: i) a pre-trained model on CIFAR-10, ii) a pre-trained model on ImageNet, and iii) a model pre-trained on ImageNet and then fine-tuned on CIFAR-10. Table 8 in our paper demonstrates the generalization ability of SampDetox to such pre-trained diffusion models.
>
> Besides, we conducted new experiments to evaluate the performance of SampDetox considering the case where pre-training datasets for diffusion models are out-of-distribution from their target classification tasks. The detailed experimental setups are as follows:
> * We conducted the training on a subset of the MS-Celeb-1M dataset. Specifically, we considered the top 100 labels with the largest number of samples and randomly selected 380 samples for each label. We split the training and tests with a ratio of 8:2 and adjusted the shape of the samples to 224*224.
> * We adopted BadNets and WaNet as attack methods, representing a visible and an invisible backdoor attack, respectively.
> * We considered four pre-trained diffusion models: i) a pre-trained model (i.e., **Model 1**) on one subset of MS-Celeb-1M, ii) a pre-trained model (i.e., **Model 2**) on another subset of MS-Celeb-1M, iii) a pre-trained model (i.e., **Model 3**) on ImageNet, and iv) **Model 4**, which is fine-tuned from Model 3 using a subset of MS-Celeb-1M.
>
> The following table shows the experimental results. We can find that SampDetox with different diffusion models achieves satisfied defensive performance on ASR. Note that, for Model 3, the CA and PA of SampDetox decrease. This is because Model 3 is pre-trained on ImageNet rather than MS-Celeb-1M, resulting in the diffusion model failing to effectively denoise MS-Celeb-1M samples. However, we can find significant improvements in the CA and PA of Model 4 since Model 4 is fine-tuned using a subset of MS-Celeb-1M. In other words, SampDetox can defend against attacks using out-of-distribution diffusion models with proper fine-tuning.​
> ​
> | Diffusion Models | &emsp;&emsp;BadNets | &emsp;&emsp;WaNet |
> | :----------------:| :------------------------: | :------------------------: |
> | | CA(%) / PA(%) / ASR(%) | CA(%) / PA(%) / ASR(%) |
> | No Defense | 94.38 / 1.27 / 99.32 | 94.61 / 1.96 / 97.05 |
> | Model 1 | 92.06 / 91.65 / 1.36 | 91.83 / 91.12 / 1.29 |
> | Model 2 | 90.27 / 89.73 / 1.31 | 89.92 / 90.05 / 1.22 |
> | Model 3 | 55.47 / 52.06 / 1.57 | 72.35 / 62.26 / 1.25 |
> | Model 4 | 88.56 / 89.27 / 1.47 | 89.43 / 89.32 / 1.30 |
>
> **AQ1:**
> Thanks for your suggestion. Please refer to **AW2** for our answer.

---

> > ### Comment · Reviewer_gGGP · 2024-08-09
> >
> > Thanks for the clarification and additional evaluation. I think the paper can be improved with a detailed explanation of the diffusion model and ablation study. Thus, I would raise my score to 6.

---

> > > ### Author Response · Authors · 2024-08-09
> > >
> > > Dear Reviewer gGGP,
> > >
> > > Thank you once again for taking the time to review our work.. We sincerely appreciate your thorough feedback and valuable insights. We will detail the diffusion model and include the rebuttal response in the final version.
> > >
> > > Best Regards,
> > >
> > > The authors of 8771

---

### Official Review · Reviewer_ok4k · 2024-07-12

**Soundness:** 4
**Presentation:** 3
**Contribution:** 3
**Rating:** 6
**Confidence:** 4

**Summary:**

This paper introduces a novel black-box backdoor purification technique consisting of two stages. In the first stage, it eliminates global perturbation by adding lightweight noise to the inputs and then leverages DDPM for denoising. Subsequently, to address visible and robust triggers, it identifies the local trigger regions, applies intense noise, and then performs diffusion denoising again.
The paper provides theoretical analysis to validate the effectiveness of its detoxification process.
In the evaluation, extensive experiments are conducted on multiple datasets, model architectures, and attacks, showing that the proposed method outperforms existing baselines. An ablation study justifies the design choices and hyper-parameters, and the paper discusses adaptive attack scenarios.

**Strengths:**

1. Observes interesting links between backdoor visibility and robustness.
2. Proposes a novel technique based on these observations.
3. Provides theoretical proof of efficacy.
4. Demonstrates support for the technique's effectiveness through extensive experiments.

**Weaknesses:**

1. Lacks evaluation of some semantic backdoors.
2. Concerns about computational overhead for deployment.
3. Evaluations of adaptive attacks are relatively weak.

**Questions:**

The paper presents an innovative concept for black-box input detoxification, highlighting noteworthy observations and techniques. However, I have several concerns I hope the authors can address:

(1) The evaluation includes 10 known backdoor attacks, which are not particularly robust. More complex backdoors utilize semantic features, for instance, [1] uses natural reflection as a trigger, and [2] employs style transferring. It would be beneficial to see evaluations against these more sophisticated backdoors.

(2) The overhead is a concern. Although Section 5.4 suggests using DDIM to reduce the overhead, it is unclear if this significantly affects the detoxification performance. Additional results on this would be helpful.

(3) The adaptive attacks discussed in Section D.2 seem relatively weak, just introducing some noise during training. A stronger attack might use natural features as triggers to enhance the robustness of BadNets. For instance, [3] concatenates two natural images as the backdoor trigger. It is suggested that the authors discuss this and potentially include stronger adaptive attacks.

Reference:

[1] Liu, Yunfei, et al. Reflection backdoor: A natural backdoor attack on deep neural networks. ECCV'20.

[2] Cheng, Siyuan, et al. Deep feature space trojan attack of neural networks by controlled detoxification. AAAI'21

[3] Lin, Junyu, et al. Composite backdoor attack for deep neural network by mixing existing benign features. CCS'20.

**Limitations:**

Please refer to the question part.

---

> ### Author Rebuttal · Authors · 2024-08-06
>
> **AQ1:**
> Thanks for your suggestion.
> Since the semantic features adopted by Reflect [1] and DSFT [2] are neither part of the original classified features nor whose distribution is the same as the distribution of the training dataset of the pre-trained diffusion model, the triggers based on such semantic features will not be restored by the denoising phase of SampDetox. Consequently, these triggers can be destroyed by SampDetox to achieve a defensive effect. To demonstrate the defense performance of SampDetox against Reflect [1] and DSFT [2], we conducted new experiments on datasets CIFAR-10, GTSRB, and Tiny-ImageNet, respectively. From the following table, we can find that SampDetox effectively defends against such backdoor attacks.
>
> | Attack Methods | &emsp;&emsp;CIFAR-10 | &emsp;&emsp;GTSRB | &emsp;Tiny-ImageNet |
> | :----------------: | :-----------------------: | :-----------------------: | :-----------------------: |
> | | CA(%) / PA(%) / ASR(%) | CA(%) / PA(%) / ASR(%) | CA(%) / PA(%) / ASR(%) |
> | Reflect | 87.46 / 88.03 / 1.72 | 93.79 / 91.43 / 2.16 | 54.52 / 51.11 / 4.10 |
> | DSFT | 86.71 / 86.55 / 1.96 | 90.45 / 89.20 / 2.05 | 50.68 / 49.85 / 4.15 |
>
>
> **AQ2:**
> Thanks for your suggestion. Since DDIM was proposed to accelerate the speed of DDPM without affecting the image generation quality, the defense performance of **SampDetox+DDIM** is comparable to that of **SampDetox+DDPM**. To demonstrate the performance of **SampDetox+DDIM**, we conducted new experiments on CIFAR-10 against ten SOTA backdoor attacks. From the following table, we can find that the defense performance of **SampDetox+DDIM** is almost the same as that of **SampDetox+DDPM**. Therefore, the utilization of DDIM to reduce the overhead will not deteriorate the detoxification performance.
>
> | Attack Methods | SampDetox + DDPM | SampDetox + DDIM |
> | :----------------: | :-----------------------: | :------------------------: |
> | | CA(%) / PA(%) / ASR(%) | CA(%) / PA(%) / ASR(%) |
> | BadNets | 89.57 / 90.15 / 2.11 | 89.49 / 90.13 / 2.12 |
> | SIG | 83.71 / 65.06 / 11.03 | 83.82 / 65.13 / 10.98 |
> | LC | 80.72 / 74.36 / 1.55 | 80.62 / 74.22 / 1.53 |
> | TrojanNN | 92.78 / 89.95 / 1.86 | 92.83 / 89.87 / 1.69 |
> | Dynamic | 88.52 / 88.62 / 1.45 | 88.52 / 88.72 / 1.42 |
> | Blended | 90.23 / 86.65 / 1.96 | 90.15 / 86.54 / 2.02 |
> | LF| 90.01 / 87.40 / 3.02 | 90.09 / 87.61 / 3.10 |
> | WaNet | 89.34 / 88.92 / 5.59 | 89.48 / 88.82 / 5.54 |
> | ISSBA | 90.74 / 86.51 / 1.60 | 90.76 / 86.65 / 1.55 |
> | BPP | 90.59 / 84.83 / 6.15 | 90.42 / 84.91 / 6.17 |
>
> **AQ3:** Thanks for your suggestion.
> As discussed in Appendix D.3 (Limitations of SampDetox), when dealing with triggers derived from semantic features that are part of the original classified features and whose distribution is the same as the distribution of the training dataset of the pre-trained diffusion model, these triggers will be restored by the denoising phase of SampDetox. In this case, the triggers will not be destroyed by SampDetox, resulting in a failure of defense.
> Since Composite Backdoor Attack (CBA) [3] concatenates two natural images as backdoor triggers, it is considered the backdoor attack type that SampDetox fails to defend.
>
> We conducted new experiments to evaluate the performance of SampDetox against the attack CBA [3] on CIFAR-10. Without defense, CA, PA, and ASR were 87.52%, 10.16%, and 90.17%, respectively. But when using SampDetox, CA, PA, and ASR were 86.81%, 38.45%, and 37.28%, respectively. The experimental results show that SampDetox has a certain effect to defend against CBA but fails to minimize the ASR.
>
> Thanks again for your suggestion. The results of new experiments in **AQ1** and **AQ3** strongly support the discussion of the limitations in Section D.3. We will add the experiments in the final version of the paper.
>
>
> **Reference**:
>
> [1] Liu, Yunfei, et al. Reflection backdoor: A natural backdoor attack on deep neural networks. ECCV'20.
>
> [2] Cheng, Siyuan, et al. Deep feature space trojan attack of neural networks by controlled detoxification. AAAI'21.
>
> [3] Lin, Junyu, et al. Composite backdoor attack for deep neural network by mixing existing benign features. CCS'20.

---

> > ### Comment · Reviewer_ok4k · 2024-08-08
> >
> > I appreciate the efforts made by the authors in addressing my concerns and questions!
> >
> > It is expected that SampDetox may fall short in defending natural triggers, e.g., composite backdoors. The additional results make sense to me and provide insights.
> > However, I'm a bit surprising that SampDetox is so effective against Reflect and DFST. It is reasonable to expect these attacks to introduce perturbations that do not align with other samples in the training set (e.g., CIFAR-10). However, this does not necessarily mean that the diffusion process overlooks these features, especially since diffusion models are typically trained on large datasets encompassing a wide range of natural images (e.g., ImageNet).
> >
> > I don't think it is quite practical to assume that the diffusion model is trained on the same dataset as the classifier due to the high computational costs involved to train a diffusion model besides the classifier.
> >
> > I suspect that the perturbation coefficient used in the image might be set too low. The coefficient, represented by $\alpha$ in the trigger injection functions of Reflect or DFST, is defined as follows:
> > x$^{'}$ = (1-$\alpha$) x + $\alpha$ x, where x is the inputs.
> > So it is likely the trigger is not robust and easily removed.
> > Would it be possible for the authors to provide illustrations of the recovered images using SampDetox for the two attacks? If the rebuttal system does not support this, then it's fine.
> >
> > Overall, the rebuttal response is well done.

---

> > > ### Author Response · Authors · 2024-08-09
> > >
> > > Dear Reviewer ok4k,
> > >
> > > We apologize for the inconvenience, but due to limitations in the rebuttal system, we are unable to provide purified samples of the two attacks in the form of images. Meanwhile, we are prohibited from providing links as per the discussion rule: ''Do not use links in your comments.'' Therefore, we attempt to describe the purified samples using words. Thank you for your understanding.
> > >
> > > * For Reflect (also called Refool), the region of trigger patterns of the purified samples is obviously destroyed compared with the original poisoned samples.
> > > * For DFST, the whole purified samples are slightly more blurred than the original poisoned samples, and the color style of purified samples is more similar to that of the original clean samples.
> > >
> > > Then, we clarify the settings of perturbation coefficients in the experiments:
> > > * Reflect generates a poisoned sample $x^{p}$ as $x^{p}= x+k \otimes x_{R}$, where $x$ is a clean sample, $x_{R}$ is a reflection image and $k$ is a convolution kernel. We optimized the value of $k$ according to the settings in its open-source code.
> > > * DFST generates a poisoned sample $x^{p}$ as $x^{p}=(1-\alpha) x+\alpha x_{s}$, where $x$ is a clean sample, $x_{s}$ is a styled image, and $\alpha$ is the perturbation coefficient. We set $\alpha$ to 0.6 following the settings in its open-source code.
> > >
> > > Additionally, to explain why SampDetox can defend against these two attacks, we conducted new experiments to calculate the robustness of their poisoned samples, respectively. For each attack, we randomly selected 50 poisoned samples on CIFAR-10 and calculated the robustness value of each sample following our robustness definition in Section 3.1 (Lines 140-146).
> > > For Reflect, the robustness of its poisoned samples has a mean value of 0.0759 and a standard deviation of 0.011. For DFST, the robustness of its poisoned samples has a mean value of 0.137 and a standard deviation of 0.019. From the experimental results, we can find that the robustness of the poisoned samples is low (ones with robustness<0.18, according to Figure 2).
> > > Therefore, when SampDetox adds enough noise to these samples, even if trigger patterns are partially restored by diffusion models, backdoors will not be successfully triggered.
> > >
> > > We will add the images of purified samples and the above experiments in the final version of the paper. We hope our response has addressed your concerns. Thanks again for your efforts in helping us improve our work.
> > >
> > > Best Regards,
> > >
> > > The authors of 8771

---

> > > > ### Comment · Reviewer_ok4k · 2024-08-09
> > > >
> > > > Thanks for the additional clarification and evaluation!
> > > >
> > > > I think they make more sense. It is likely the poisoned models of Refool or DFST pay a lot attention on some low-level features (although they look natural) that are not robust to the input purification.
> > > >
> > > > I increase my rating and hope the authors could properly include the rebuttal reponse to their final version.

---

> > > > > ### Author Response · Authors · 2024-08-09
> > > > >
> > > > > Dear Reviewer ok4k,
> > > > >
> > > > > Thank you once again for your efforts in helping us improve our work. We sincerely appreciate your thorough feedback and valuable insights. The rebuttal response will be included in the final version of the paper.
> > > > >
> > > > > Best Regards,
> > > > >
> > > > > The authors of 8771

---

### Official Review · Reviewer_aKFd · 2024-07-13

**Soundness:** 3
**Presentation:** 3
**Contribution:** 3
**Rating:** 7
**Confidence:** 4

**Summary:**

This paper proposes SampDetox, a novel black-box backdoor defense method using perturbation-based sample detoxification. The key contributions are:

1. A preliminary study revealing the correlation between trigger visibility and poisoned sample robustness.
2. A two-stage defense approach combining lightweight global noise and targeted local noise to destroy both visible and invisible backdoor triggers.
3. Use of diffusion models to denoise and restore samples after perturbation.
4. Comprehensive experiments demonstrating effectiveness against various state-of-the-art backdoor attacks.

**Strengths:**

1. Novel approach combining noise perturbation and diffusion models for backdoor defense
2. Comprehensive evaluation against 10 different backdoor attacks
3. Theoretical analysis supporting the approach (Theorems 4.1 and 4.2)
4. Does not require access to model parameters or training data, making it applicable to MLaaS scenarios (black-box models)
5. Compared to previous work, it addresses both visible and invisible backdoor triggers

**Weaknesses:**

1. The effectiveness of SampDetox heavily depends on the correlation between trigger visibility and robustness, but this relationship may not always hold for more sophisticated attacks.
2. The two-stage approach with fixed noise levels (t1 and t2) may be too rigid to adapt to diverse trigger patterns, potentially missing subtle backdoors or over-perturbing clean regions.
3. The approach may remove benign high-frequency details from images, potentially affecting classification accuracy for tasks that rely on such fine-grained features.

**Questions:**

1. The paper suggests optimal values for t1 and t2 based on your experiments. How sensitive is the performance of SampDetox to these parameters across different datasets or attack types? Is there a way to automatically determine optimal values for a given scenario?
2. How does the pixel-wise SSIM calculation in Stage 2 scale with image size? Is there a more efficient way to locate high-visibility triggers without compromising effectiveness? Moreover, how does the method perform on very large images or high-resolution datasets? Is there a practical upper limit on image size due to computational constraints?

**Limitations:**

The authors have adequately addressed the limitations.

---

> ### Author Rebuttal · Authors · 2024-08-06
>
> **AW1:**
> Indeed, the effectiveness of SampDetox depends on our observed correlation. However, this does not mean that our approach cannot prevent more sophisticated attacks. This is because our observation has good generalization ability, since our observation is based on ten SOTA backdoor attacks covering most mainstream backdoor attack types. In other words, it can reflect the commonality of most attacks.
> In addition, in Appendix D.2, we tried to design adaptive attacks with high robustness and low visibility that aim to break the observed correlation. However, the designed adaptive attacks cannot achieve high enough robustness while keeping low visibility, which can be successfully defended by SampDetox. To the best of our knowledge, it is still challenging to propose an effective attack with high robustness and low visibility, which is worthy of further study.
>
> **AW2:**
> Sorry for the confusion. Although the value of $\overline{t}_2$ is fixed, the noise levels imposed on sample pixels are not fixed. Please note the hyperparameter $\overline{t}_2$ in Stage 2 represents the maximum added noise level of each pixel rather than its actual added noise level. Based on both the original sample and its denoised version obtained in Stage 1, SampDetox calculates different noise levels for different pixels in Stage 2 (Line 9 of Algorithm 1), which aims to adapt to various trigger patterns. From Tables 1, 3, 4, 5, and 6, we can find the superiority of SampDetox in defending against various attacks involving different datasets and models.
>
> **AW3:**
> Indeed, our approach may remove benign high-frequency. However, this does not eclipse the effectiveness of our approach against backdoor attacks. To evaluate the defense performance of SampDetox on classification tasks that rely on benign high-frequency details, we conducted new experiments on the DTD (Describable Textures Dataset) dataset, where we adopted a visible backdoor attack, i.e., BadNets, and an invisible backdoor attack, i.e., WaNet, respectively. The following table compares SampDetox with six baselines. From this table, we can find that though the CA of SampDetox a little smaller than the CA of ``No Defense’’, SampDetox can achieve the best defense performance on CA, PA, and ASR compared with all baselines, which shows the applicability and superiority of SampDetox.
>
>
> | Defense | BadNets | WaNet |
> | :--------------: | :----------: | :-------------------: |
> | | CA(%) / PA(%) / ASR(%) | CA(%) / PA(%) / ASR(%) |
> | No Defense | 65.42 / 2.05 / 90.56 | 64.72 / 2.17 / 88.56 |
> | Sancdifi | 48.98 / 51.23 / 2.84 | 49.64 / 28.06 / 15.12 |
> | BDMAE | 56.25 / 56.43 / 1.86 | 55.57 / 19.17 / 47.22 |
> | ZIP | 43.17 / 40.95 / 3.32 | 44.87 / 40.56 / 3.31 |
> | Noising | 37.55 / 35.87 / 5.26 | 38.46 / 37.08 / 3.27 |
> | ShrinkPad | 41.27 / 39.36 / 8.92 | 43.19 / 42.48 / 2.92 |
> | Blur | 33.17 / 30.92 / 12.05 | 33.49 / 29.50 / 3.67 |
> | SampDetox (Ours) | **57.46** / **57.12** / **1.73** | **58.33** / **58.12** / **2.21** |
>
> **AQ1:**
> Sorry for your confusion. We use the same values of $\overline{t}_1$ and $\overline{t}_2$ throughout the paper. In other words, utilizing fixed $\overline{t}_1=20$ and $\overline{t}_2=120$, SampDetox can defend against various types of attacks involving different datasets and models, as shown in Tables 1, 3, 4, 5, and 6.
> In general, given a new scenario, there is no need to change the value of $\overline{t}_1$ and $\overline{t}_2$ to achieve satisfactory defense performance.
>
> **AQ2:**
> Sorry for your confusion. For two $n * n$ images A and B, we calculate their Pixel-SSIM with a block size of $t*t$ as follows:
> $\operatorname{Pixel-SSIM}(x, y)=\frac{\left(2 \mu_x \mu_y+c_1\right)\left(2 \sigma_{x y}+c_2\right)}{\left(\mu_x^2+\mu_y^2+c_1\right)\left(\sigma_x^2+\sigma_y^2+c_2\right)}$, where $x$ and $y$ are the same positional pixels of images A and B, respectively. $\mu_x$ and $\sigma_x$ are the mean and variance of the block centered on $x$, respectively. $\sigma_{xy}$ is the covariance between $block_x$ and $block_{y}$, and $c_1$ and $c_2$ are constants. According to the above formula, the time complexity for the pixel pair is $O(t^2)$. Thus the time complexity of whole Pixel-SSIM is $O(t^2 n^2)$. Considering that $t$ is constant in practice, the computational cost of Pixel-SSIM is determined by the number of pixels. In addition, we conducted new experiments to calculate the overhead of Pixel-SSIM. The experimental results show that the Pixel-SSIM overhead only accounts for less than 1% of the overall overhead of SampDetox. In other words, such overhead can be neglected in practice.
>
> We conducted new experiments on the ImageNette dataset (a subset of 10 classes from ImageNet) containing larger images with a resolution of $256*256$. From the following table, we can find that SampDetox is still effective in defending against attacks on large images.
>
> The practical upper limit of image size depends on the denoising capability of diffusion models rather than the computational constraints. When the size of images is too large, it is difficult to train an effective diffusion model to denoise such images. Nevertheless, the experimental results show that SampDetox is still effective in common image sizes (i.e., 256*256).
>
> | Attack Methods | No Defense | SampDetox |
> | :------------: | :--------------------: | :--------------------: |
> | | CA(%) / PA(%) / ASR(%) | CA(%) / PA(%) / ASR(%) |
> | BadNets | 83.75 / 10.83 / 95.09 | 79.59 / 79.74 / 1.91 |
> | SIG | 77.45 / 10.09 / 96.26 | 75.15 / 74.85 / 6.43 |
> | LC | 76.38 / 10.46 / 90.24 | 73.57 / 73.13 / 1.92 |
> | TrojanNN | 84.61 / 10.91 / 98.86 | 83.04 / 83.11 / 1.66 |
> | Dynamic | 83.73 / 11.58 / 97.02 | 81.16 / 80.63 / 1.64 |
> | Blended | 84.39 / 10.66 / 98.22 | 83.02 / 82.49 / 1.08 |
> | LF | 85.19 / 11.39 / 92.65 | 82.15 / 81.95 / 2.51 |
> | WaNet | 83.34 / 9.49 / 87.59 | 82.16 / 82.22 / 4.23 |
> | ISSBA | 85.26 / 11.49 / 90.37 | 82.88 / 82.45 / 2.33 |
> | BPP | 83.04 / 10.30 / 91.11 | 82.23 / 81.93 / 3.55 |

---

> > ### Comment · Reviewer_aKFd · 2024-08-12
> >
> > Thank the authors for their detailed rebuttal. All my concerns have been addressed, so I am raising my score accordingly.

---

> > > ### Author Response · Authors · 2024-08-13
> > >
> > > Dear Reviewer aKFd,
> > >
> > > Thank you once again for taking the time to review our work. We sincerely appreciate your thorough feedback and valuable insights. The rebuttal response will be included in the final version of the paper.
> > >
> > > Best Regards,
> > >
> > > The authors of 8771

---

### Decision · Program_Chairs · 2024-09-25

**Decision:**

Accept (poster)

**Comment:**

This work proposes a black-box approach to defend against backdoor attacks, based on a diffusion model that restores the image parts covered by the backdoor trigger. Most of the concerns raised by the reviewers were addressed, even if the paper still falls short in evaluating against adaptive attacks and discussing the limitations of the approach, which (i) strongly relies upon the effectiveness of the pre-trained diffusion model, and (ii) remains specific to image data. Nevertheless, the authors can better discuss and highlight such limitations, while the paper does a good job when it comes to the experimental analysis. Overall, also in light of the positive feedback from the reviewers after the rebuttal, I can recommend acceptance.